# Aerosol-induced intensification of cooling effect of clouds during Indian summer monsoon

Chandan Sarangi [1,2], Vijay P. Kanawade [3], Sachchida N. Tripathi[1], Abin Thomas[3] & Dilip Ganguly[4]

Measurements and models show that enhanced aerosol concentrations can modify macro- and micro-physical properties of clouds. Here, we examine the effect of aerosols on continental mesoscale convective cloud systems during the Indian summer monsoon and find that these aerosol–cloud interactions have a net cooling effect at the surface and the top-of-atmosphere. Long-term (2002–2016) satellite data provide evidence of aerosol-induced cloud invigoration effect (AIvE) during the Indian summer monsoon. The AIvE leads to enhanced formation of thicker stratiform anvil clouds at higher altitudes. These AIvE-induced stratiform anvil clouds are also relatively brighter because of the presence of smaller sized ice particles. As a result, AIvE-induced increase in shortwave cloud radiative forcing is much larger than longwave cloud radiative forcing leading to the intensified net cooling effect of clouds over the Indian summer monsoon region. Such aerosol-induced cooling could subsequently decrease the surface diurnal temperature range and have significant feedbacks on lower tropospheric turbulence in a warmer and polluted future scenario.

[1] Department of Civil Engineering and Centre for Environmental Science and Engineering, Indian Institute of Technology Kanpur, Kanpur 208016 UttarPradesh, India. [2] Pacific Northwest National Laboratory, Richland, WA 99352, USA. [3] Centre for Earth, Ocean & Atmospheric Sciences, University of Hyderabad, Hyderabad, Telangana 500046, India. [4] Centre for Atmospheric Sciences, Indian Institute of Technology Delhi, New Delhi 110016, India. Correspondence and requests for materials should be addressed to V.P.K. (email: vijaykanawade03@yahoo.co.in) or to S.N.T. (email: snt@iitk.ac.in)

Earth's radiative balance is strongly modulated by the presence of clouds. Atmospheric aerosols can affect cloud formation processes via activation of cloud condensation nuclei (CCN), thereby modifying the cloud radiative forcing (CRF) as well as precipitation patterns. This aerosol-mediated change in CRF is termed as the aerosol indirect effect (AIE)[1–3], which constitutes the largest uncertainty in the current climate forcing[4]. This ambiguity in AIE is primarily due to the regional variability in aerosols, clouds, and the complexity in aerosol-related changes in dynamical feedbacks at different spatio-temporal scales. Moreover, the influence of environmental conditions and contingent surface feedbacks on aerosol–cloud interactions is also a considerable source of uncertainty in the estimation of AIE[5–7]. While AIE is reasonably well understood for marine warm clouds, it remains elusive for continental mixed-phase clouds associated with mesoscale convective systems (MCSs)[8,9]. Thus, cloud-regime and region-specific investigations of aerosol–cloud forcing response are required to advance the current process-level understanding of aerosol–cloud–radiation–climate interactions[10,11].

Generally, polluted clouds under high aerosol loading consist of more number of smaller cloud droplets[2], which increases the cloud reflectance or brightness, thereby causes the cooling effect of the Earth[12]. The more number of smaller cloud droplets reduces the efficiency of droplet growth by collision-coalescence, which inhibits early rain formation and increases cloud lifetime[1,13]. Increase in cloud lifetime can reflect more incoming solar radiation inducing a net cooling effect. However, under convective conditions, such delay in the rain formation simultaneously increases droplet mobility[14], leading to more water mass aloft in the atmosphere. Ice formation processes within the ascending cloud begin once the freezing level is reached and the more latent heat freezing released further invigorates the vertical development of the cloud. This phenomenon is referred to as the aerosol invigoration effect (AIvE), which was initially noted by Williams et al.[15] followed by many studies over continental as well as ocean surfaces[16–23]. Cloud-resolving modeling studies further illustrated that AIvE is primarily associated with the increase in cloud top height (CTH), cloud fraction (CF), vertical lifting of cloud droplets followed by a subsequent increase in ice-phase hydrometeors[24–33]. The ensuing processes in AIvE are relatively better understood under both continental and ocean conditions[19,34], but only a few studies have investigated the AIvE-associated CRF and related surface feedbacks under continental conditions[7,9]

For a developing deep convective cloud (DCC) system, the AIvE on shortwave (SW) CRF is always negative because of the persistent enhancement of the cloud albedo. However, the cloud albedo begins to saturate as the DCCs reach a mature stage. Thus, AIvE-induced cooling in dissipating DCCs is rather insignificant. But, the AIvE-induced deep clouds with cold cloud tops emit less thermal radiation to the space. This induces large positive longwave (LW) CRF at the top-of-atmosphere (LWCRF$_{TOA}$), which may cause a net warming effect of AIvE on MCSs[28,34,35]. In contrast, AIvE-associated increase in occurrence and lifetime of stratiform anvil cloud branch of mature MCSs and microphysical changes such as reduction in the size of ice particles can also significantly enhance the cloud albedo, thus causing a net cooling at the TOA and the surface[29,36]. Thus, the overall net radiative effect associated with the AIvE is a function of both macro- and micro-physical properties of the prevalent MCSs[34].

In this study, we investigate the aerosol-induced CRF for the mesoscale mixed-phase convective clouds system developed under continental conditions during the highly dynamic Indian Summer Monsoon (ISM). Prevalence of low-pressure system over India is associated with north-westward propagation of associated MCSs across ISM region. Moisture advection, the formation of numerous convective cells and heavy rainfall progress from the Indian peninsula to the foothills of Himalaya during this period. While inter-annual variability in aerosol–cloud associations over India is identified[37], recent studies have indicated the predominance of AIvE over ISM region[19,38–43]. But, the associated aerosol–cloud–radiative effect is currently unknown[9,44]. ISM contributes to more than 90% of the total annual rainfall over India[45] and plays an important role in India's agriculture and economy. During mid-June through October ISM prevails over continental central India (about 70% of total ISM rainfall)[45] and is identified as one of the extreme water scarcity zones globally[46]. Hence, the significance of aerosol-induced CRF and its feedback on surface forcing over this region of climatic and hydrological importance cannot be overemphasized.

## Results

**Cloud physical, optical and radiative properties**. This study focuses on ISM region (17°N–27°N and 74°E–88°E, hereafter referred to as the ISMReg shown by the bounded box in Supplementary Figure 1A). Supplementary Table 1 briefly summarizes all datasets used and their corresponding temporal resolution applied. The frequency distribution of different cloud types on cloud optical thickness (COT)–cloud top pressure (CTP) axes (Fig. 1a) illustrates that about 70% of clouds formed over the ISMReg are thick (high COT) and deep (CTP < 500 hPa) indicative of the dominant mesoscale deep convective system. This has previously been well established by satellite-based Lidar (CALIPSO) and radar (CloudSat) observations[47–49]. The instantaneous SWCRF$_{TOA}$, LWCRF$_{TOA}$, NETCRF$_{TOA}$, all-sky albedo ($A_{all-sky}$) and CF on similar COT–CTP axes (Fig. 1b–f, respectively) further illustrate that these DCCs are associated with relatively low positive LWCRF$_{TOA}$ (due to taller clouds) and high negative SWCRF$_{TOA}$ (due to high CF, $A_{all-sky}$ and COT) leading to high instantaneous net cooling over this region (NETCRF$_{TOA}$ of about −200 W/m²). Supplementary Figure 1 depicts the climatological mean (2002–2016) of Moderate Resolution Imaging Spectroradiometer (MODIS) aerosol optical depth (AOD) and the diurnal mean of NETCRF$_{TOA}$. On an average, the daily mean NETCRF$_{TOA}$ over the ISMReg also ranged from about −20 to −40 W/m² indicative of the net cooling effect of monsoon clouds. Despite heavy rainfall (climatological mean annual rainfall of ~700 mm), the ISMReg has high aerosol loading (AOD > 0.4) during the summer monsoon period, perhaps due to the persistent surface emissions of anthropogenic aerosols at high rates[50]. Thus, the combined presence of aerosols and clouds over this region creates a unique natural laboratory for investigating aerosol–cloud–radiation interactions over continents.

**Aerosol-cloud-radiative forcing associations**. Following the early work on aerosol-induced cloud brightening effect by Twomey and Warner[51], the fundamental investigations have linked to an increase in aerosol concentrations with variation in all-sky albedo and CRF via micro-physical changes in cloud droplet distribution for similar clouds. In Fig. 2, the collocated measurements of aerosol, cloud properties and CRF based on 1° × 1° gridded dataset within ISMRreg are first identified. Then, the MODIS observed AOD values are sorted in ascending order and divided into 50 equal bins (i.e. each bin of 2 percentiles). The collocated AOD and cloud properties (CF, CTP and $A_{All-Sky}$) and CRFs (NETCRF, LWCRF and SWCRF) at the TOA and the surface for each of these 50 bins were then averaged and plotted (Fig. 2a–e). A remarkable linear relationship between CF and $A_{All-Sky}$ is evident (Fig. 2a). A similar linear association between CF and CTP is also observed (Fig. 2b). This illustrates that $A_{All-Sky}$

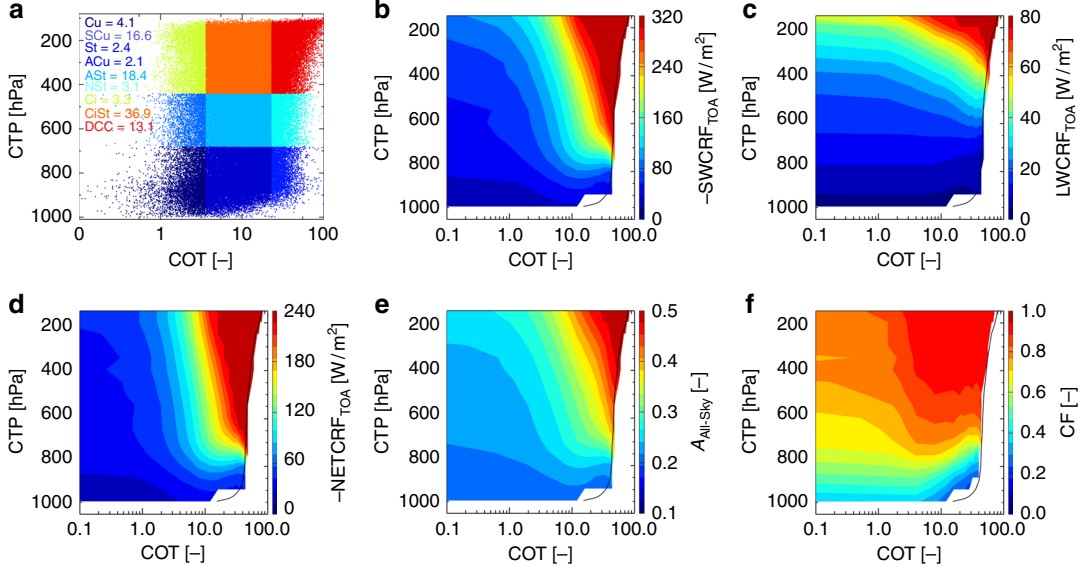

**Fig. 1** Climatology of occurrence of different cloud types, and their physical, optical and radiative properties. **a** The frequency distribution plot for cloud types over the ISMReg (shown by the bounded black box in Supplementary Figure 1A). Cloud types identified from the International Satellite Cloud Climatology Project (ISCCP) definition. **b–f** Contour plots of COT versus CTP as a function of **b** $SWCRF_{TOA}$, **c** $LWCRF_{TOA}$, **d** $NETCRF_{TOA}$, **e** $A_{All-Sky}$ and **f** CF. The solid black line in **b–f** indicates grid pixels with an average of 30 data points, i.e. all color pixels to the left of the solid line is the average of greater than 30 data points

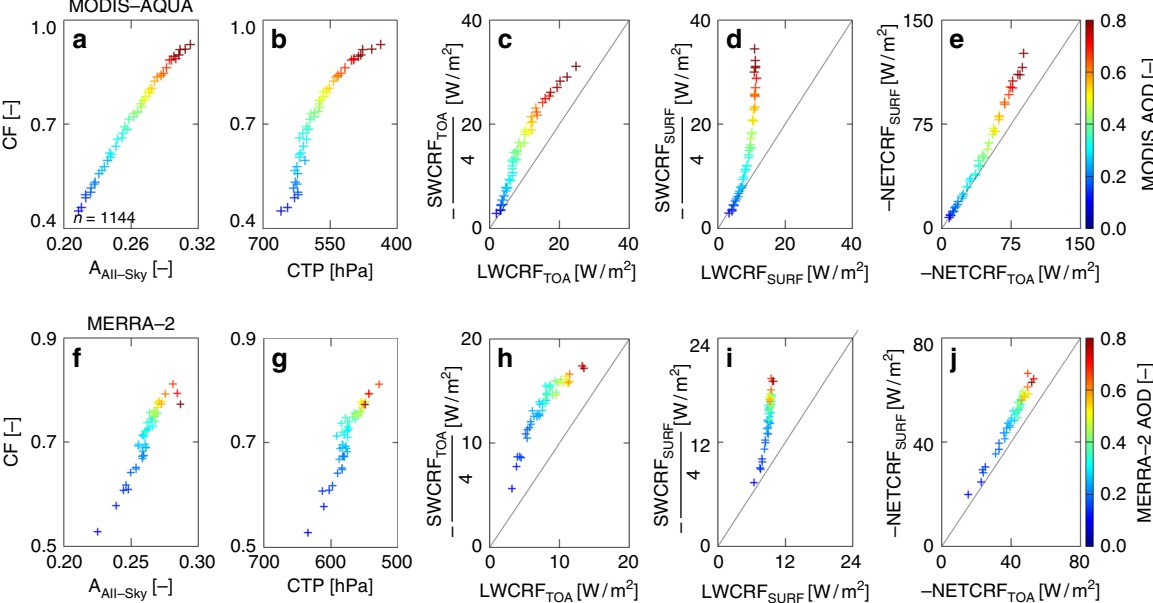

**Fig. 2** Cloud physical and radiative properties as a function of aerosol optical depth. Scatter plots between **a** $A_{All-Sky}$ and CF, **b** CTP and CF, **c** $LWCRF_{TOA}$ and $SWCRF_{TOA}$, **d** $LWCRF_{SURF}$ and $SWCRF_{SURF}$, and **e** $NETCRF_{TOA}$ and $NETCRF_{SURF}$, respectively, as a function of MODIS AOD and scatter plots between **f** $A_{All-Sky}$ and CF, **g** CTP and CF, **h** $LWCRF_{TOA}$ and $SWCRF_{TOA}$, **i** $LWCRF_{SURF}$ and $SWCRF_{SURF}$, and **j** $NETCRF_{TOA}$ and $NETCRF_{SURF}$, respectively, as a function of MERRA-2 AOD. Y-axis in panels **c**, **d** and **h**, **i** are downscaled by a factor of ¼. The diagonal lines in **c–e** and **h–j** indicate a 1:1 line. Each scatter point (plus sign) in all panels (**a–j**) is average of "n" number of data samples ($n = 1144$)

and CTP are a positive function of increasing AOD. Distinct separation of color with an increase in AOD is clearly visible, even at high CF (0.8–0.9) conditions, indicating a negligible saturation of these associations. For instance, an increase in AOD from 20 to 80 percentiles is associated with an increase in albedo of about 0.1 (Fig. 2a) and enhancement in CTP of about 100 hPa (Fig. 2b). Cloud albedo ($A_{cld}$), CF and $A_{All-Sky}$ are related by relation: $A_{All-Sky} = CF \times A_{cld} + (1 - CF) \times A_{clr}$[52], where $A_{clr}$ is the clear-sky albedo. The linearity between CF and $A_{All-Sky}$, as

observed in Fig. 2a, indicates distinctiveness between CF and $A_{cld}$. Thus, AOD-associated increase in both intrinsic (cloud albedo) and extrinsic (CF) forcing is observed simultaneously, yet independently, over the ISMReg. The combined increase in CF, CTP and $A_{All-Sky}$ with aerosol loading suggests the occurrence of AIvE over the ISMReg. Moreover, macro-level relationships among $A_{All-Sky}$ (or CRF), CF and AOD are more robust and generally have less ambiguity compared to poorly constrained micro-level associations[37]. This distinct and robust positive association

between AOD–CF–CTP and $A_{cld}$ can be attributed to AIvE effect[19], mainly due to the presence of aerosol-associated micro-physical changes in MODIS and CloudSat observations over the ISMReg (Supplementary Fig. 2). Aerosol-induced cloud invigoration over the IMSReg is further substantiated and presented in the Supplementary Note 1.

Figure 2c shows AOD-associated increase in $LWCRF_{TOA}$ with a corresponding increase in $SWCRF_{TOA}$. In general, an increase in aerosol loading from 20 to 80 percentile is associated with a large increase in $SWCRF_{TOA}$ (~80 W/m$^2$) than $LWCRF_{TOA}$ (~10 W/m$^2$). However, under the highly polluted scenario the gradient between $SWCRF_{TOA}$ and $LWCRF_{TOA}$ is comparatively lower. This is probably because of the difference in AOD-CTP gradient between low CF and high CF conditions. While the rate of AOD-associated increase in $A_{All-Sky}$ remains same for the entire CF range (Fig. 2a), the aerosol-associated increase in CTP is comparatively higher at high CF compared to a low CF scenario (Fig. 2b). This implies that the net cooling effect by aerosols is canceling the net warming effect from AOD-induced increase in CTH over the ISMReg under highly polluted conditions.

Similarly, aerosol-associated enhancement in $SWCRF_{SURF}$ is much higher than that of $LWCRF_{SURF}$ (Fig. 2d). This means that high aerosol loading is associated with high CF and CTP over the ISMReg resulting in significant attenuation of incoming solar radiation reaching at the surface. For instance, with the increase in AOD from 20 to 80 percentile, $SWCRF_{SURF}$ and $LWCRF_{SURF}$ increases by ~80 and ~5 W/m$^2$, respectively, this results in a net cooling effect at the surface. Thus, irrespective of the location or phase of the cloud, increase in aerosol loading within the ISMReg is associated with an intensification of cooling effect of clouds at both the TOA and the surface due to increase in the magnitude of SWCRF. Further, comparison of $NETCRF_{TOA}$ and $NETCRF_{SURF}$ (Fig. 2e) illustrates that an increase in AOD from 20 to 80 percentile is associated with the enhanced cooling effect of clouds at the surface (~75 W/m$^2$) compared to that at the TOA (~60 W/m$^2$). The higher cooling at the surface compared to that at the TOA suggests aerosol-associated warming of Earth's atmosphere via aerosol micro-physical effect.

**Causality.** MODIS-retrieved AOD values can be contaminated under cloudy conditions[53]. Therefore, to further corroborate and enhance the reliability of our findings, we repeated this analysis (Fig. 2a–e) using Modern-Era Retrospective analysis for Research and Applications Version-2 (MERRA-2) reanalysis AOD dataset[54,55] (Fig. 2f–j). MERRA-2 AOD dataset is not cloud contaminated, and thus unaffected by cloud contamination[56]. Interestingly, a very similar association between MERRA-2 AOD and various cloud properties is observed (Fig. 2f–j). The variability in the coarsely resolved MERRA-2 aerosol product is found to be smaller than satellite measurements. A very good spatial correlation between MODIS and MERRA-2 AOD is also observed over India (Supplementary Figure 3). Supplementary Figure 4 further shows similar aerosol–cloud properties associations using both MODIS and MERRA-2 AOD. This indicates that AOD–albedo–CRF positive associations observed in Fig. 2 are indeed robust and unlikely to be manifested by cloud contamination of AOD data over the ISMReg.

In addition, the positive association between cloud properties and AOD can also be due to meteorological co-variability under cloudy conditions[57]. We have performed multiple linear regression analysis to test the co-variability of aerosol loading and meteorological parameters (e.g. temperature, relative humidity, wind shear, and geopotential height) on the CF, SWCRF, LWCRF, NETCRF and $A_{All-sky}$. The meteorological parameters are obtained from NOAA-NCEP Global Data Assimilation

System (GDAS). We found that the regressed slopes between MODIS AOD, and the cloud properties are much higher compared to that due to the meteorological variability (Supplementary Table 2). This suggests that variability in cloud physical and optical properties are closely associated with variability in AOD and not by meteorological variability. Moreover, a recent study made an exhaustive examination of the role of aerosol humidification effect on aerosol–cloud associations using narrowed RH regimes and radiosondes over the ISMReg to show that aerosol humidification effect has a negligible contribution to the observed positive associations in AOD-albedo-SWCRF over ISMReg[19].

**Cloud-resolving model simulations.** To further corroborate our observational findings in Fig. 2, two realistic aerosol-sensitive numerical experiments on Weather Research and Forecasting (WRFv3.5.1) platform at cloud-resolving scale (3 km) are also performed over the ISMReg. We used aerosol-aware Thompson micro-physical parameterization which accounts for aerosol micro-physical effect by prescribing emission rates of CCN at the surface according to the Goddard Chemistry Aerosol Radiation and Transport (GOCART) climatology. Details about the parameterization and CCN prescriptions are provided in Thompson and Eidhammer[58]. A typical low depression period over the ISMReg during 12–17th August 2011 is simulated at cloud-resolving scale with default present-day emission rates of CCN (high CCN run). Another simulation with a very low CCN scenario (low CCN run; emission rates of 100 times lower than that of the default run) is also performed. Thus, the differences in physical and optical properties of the simulated MCSs between these two runs can be solely attributed to CCN-induced micro-physical and dynamical changes to MCSs. The details about the model setup, evaluation of synoptic meteorology and cloud properties are provided in the Methods section and Supplementary Figs. 5–8.

In accordance with the above discussion, the difference between high and low CCN simulations illustrate that increase in CCN concentration causes an increase in mean columnar liquid water content (cloud + rain), mean columnar ice-phase water content (IWC) (Supplementary Figure 9A, B) as well as diurnally averaged mean SWCRF, LWCRF and NETCRF at both the TOA and the surface (Supplementary Figure 9C–H). Differences in domain-averaged mean vertical profiles also show large enhancement in IWC associated with strong updrafts in high CCN case, mainly above 4 km altitude (i.e. freezing level) (Supplementary Figure 10A, B). CCN-induced increase in liquid water content above the freezing level is also evident, suggesting an upward shift of smaller cloud drops in high CCN loading, which persists as supercooled liquid drops above the freezing level. These supercooled cloud droplets above the freezing level results in the enhanced formation of ice hydrometeors and thereby releases a higher amount of latent heat above the freezing level, which further intensifies the updrafts. This chain of processes results in an increase of overall cloudiness, cloud depth and higher CRF values under high aerosol loading. Thus, these numerical simulations strengthen our observational findings (Fig. 2) and support the hypothesis that AIvE of MCSs prevalent during ISM induces a net cooling effect at the TOA and the surface (rather than a mere-meteorological correlation).

**Analysis of AIvE on convective and stratiform anvil clouds.** For a detailed understanding, the AIvE-induced changes in convective tower clouds (TCs) and stratiform anvil clouds (SACs) separately, and their contribution to the overall observed AOD–CRF associations are investigated. Following Koren et al.[59], MODIS-

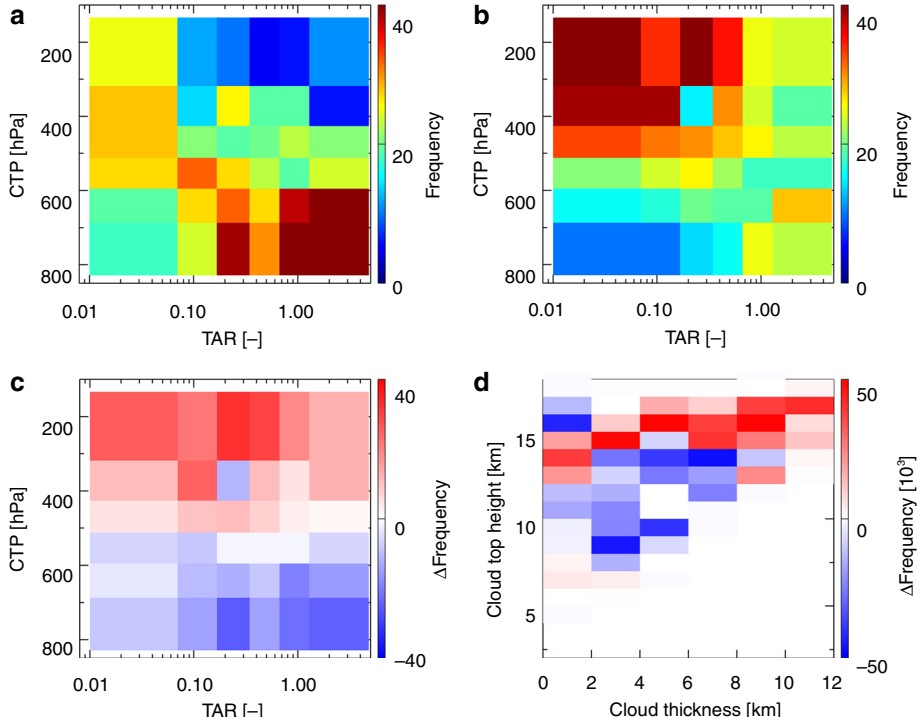

**Fig. 3** Variation in occurrence of tower and anvil clouds under low and high aerosol loading scenarios. Contour plot of TAR versus CTP as a function of cloud frequency for **a** low AOD, **b** high AOD, and **c** difference (high–low AOD). **d** WRF model simulated difference (high–low aerosol) in the cloud frequency as a function of cloud thickness and cloud top height for the deep convection clouds

retrieved Level-2 $COT_{ICE}$ in each $1° × 1°$ grid is used to calculate the observed number of tower ($N_T$: $COT_{ICE} > 10$) and anvil ($N_A$: $COT_{ICE} < 10$) cloud pixels (refer to the Methods section). Daily tower-to-anvil ratio (TAR = $N_T/N_A$) within each $1° × 1°$ grid over the ISMReg (collocated data point to all other datasets) is calculated. Further, all collocated cloud, AOD and radiation data are segregated into two bins: high AOD scenario (AOD > 66 percentiles) and low AOD scenario (AOD < 33 percentiles), and OLR < 240 W/m² is used in the analysis to represent deep convective cloud conditions. The cloud occurrence frequency distribution for low (Fig. 3a) and high (Fig. 3b) AOD scenarios and difference in high and low AOD scenarios (Fig. 3c) are plotted on CTP–TAR space. The TAR values are arranged in ascending order and divided into 6 bins of 18 percentiles each, discarding 2 percentiles from either end to avoid extreme values. CTP bins are also formed a similar way to create 36 equally populated data sample groups in the CTP–TAR space. The CTP–TAR plots synergistically illustrate TCs (TAR > 1) and the SACs (TAR < 1). It is apparent that the dominance of SACs increases as we move from right to left in the CTP–TAR space. For a low aerosol loading scenario, the peak frequency of occurrence of CTP for TCs is about 450 hPa, and the same for SACs increases from CTP of about 400 hPa (for TAR of ~0.5) to CTP of about 250 hPa (for TAR of ~0.05). But, for high aerosol loading scenario, the peak frequency of occurrence of TCs as well as SACs is at higher altitude (CTP of about 250 hPa). Further, the difference between these scenarios clearly illustrates that the occurrence of TCs and SACs increases at all altitudes above 500 hPa. Interestingly, the increase in the frequency of SACs is greater than that inTCs under high aerosol loading. In general, most of the TCs grow and mature upto ~400 hPa, then it tends to grow vertically as they transform into SACs in the low aerosol loading scenario, perhaps due to radiative differences at the top and bottom of SACs[60]. In contrast, for high aerosol loading condition, most of the TCs fully

develop into DCCs due to AIvE and advect out as stratiform/anvil clouds because of the thermal capping at the tropopause and/or advective force from the tropical easterly jet (TEJ) at altitudes above 300 hPa (nearly 10 km) during ISM[61,62].

In this framework, we also used our model simulations to compare the AIvE-induced changes in cloud distribution between low and high CCN simulations. Simulated CF profile at hourly resolution is used over each grid of the model to locate the cloud base height, CTH and cloud thickness (CT) of various cloud layers. A cloud layer is defined as a continuous stretch of finite CF values which exceeds more than 2 km. There may be more than one cloud layers over a grid column, but we have considered the topmost cloud layer to be consistent with MODIS CTP observations. The altitude of the model levels corresponding to cloud top and cloud base of each cloud layer is stored. CT of each cloud layer is calculated by subtracting cloud base height from the CTH. The aerosol-induced (high–low CCN) changes in cloud distribution (Fig. 3d) is plotted on CTH–CT space. The clouds lying close to the diagonal line connecting the left bottom and right top corner on CTH–CT axes are the growing TCs (as their CT increase almost linearly with CTH) whereas the clouds in the right top quadrant (high CTH and high CT) represent fully developed TCs. The SACs are represented by clouds with high CTH (~12–16 km), but CT less than 4–5 km.

The simulated cloud distribution is dominated by the presence of thick cloud layers (9–11 km) extending up to 12–16 km altitude. In addition, numerous cloud layers with CTH at 12–16 km and CT of about 3–7 km are simulated. Thus, the case study analyzed here represents well developed TCs and associated SACs over the ISMReg (Supplementary Figure 11A, B). The aerosol-induced differences in the cloud occurrence frequency (Fig. 3d) clearly indicate that the clouds with deeper CTH and larger CT increased significantly under high CCN case indicative of the AIvE. Also, for clouds with CTH > 13 km, the increased amount

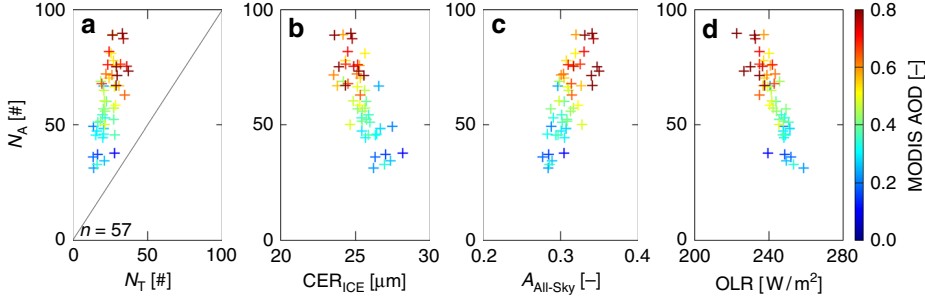

**Fig. 4** Stratiform anvil and tower clouds associated change in cloud micro-physical and radiative properties as a function of aerosol optical depth. Associations of **a** $N_T$, **b** $CER_{ICE}$, **c** $A_{All-Sky}$ and **d** OLR with $N_A$ as a function of MODIS AOD. Each scatter point (plus sign) in all panels (**a–d**) is the average of "n" number of data samples ($n = 57$)

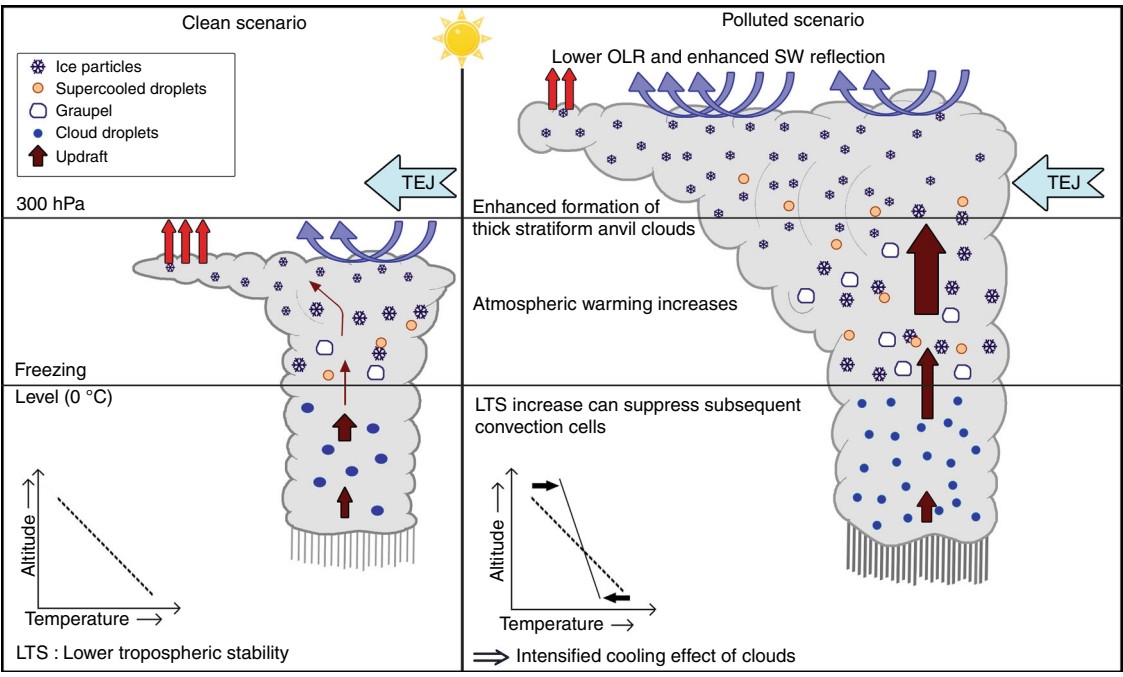

**Fig. 5** Processes leading to intensified cooling effect of clouds under polluted scenario over the Indian summer monsoon region

of thicker SACs, but decreased thinner anvil clouds (CT < 4 km), suggesting the formation of thicker SACs clouds under polluted condition. It is worth mentioning that about 20 thousand cloud layers are used in this analysis, which underlines the statistical robustness of these simulated differences. While the environmental condition can impact the magnitude of the aerosol-induced enhancement in SACs, the fact that our model comparison is in remarkably good agreement with both MODIS and MERRA-2 analysis, strongly reinforces the seminal role of AIvE on cloud structure and distribution in MCSs over the ISMReg.

Further, the associations of $N_A$–$N_T$, $N_A$–$CER_{ICE}$, $N_A$–$A_{All-Sky}$ and $N_A$–OLR as a function of AOD are studied (Fig. 4). Here, 50 scatter points are created using the same methodology as in Fig. 2. First, as expected, a linear increase in $N_A$ and $N_T$ is observed with an increase in aerosol loading. The ratio of $N_A$ and $N_T$ is nearly 1 (the bottom left corner) for low aerosol loading, but the distribution is heavily skewed towards $N_A$ under high aerosol loading suggesting the large increase in $N_A$ compared to $N_T$. $N_A$ ($N_T$) increased from ~35 (21) to ~76 (36) as AOD increased from 0.25 to 0.75. This emphasizes that AIvE cause significant enhancement of SACs. Interestingly, it is also seen that the size of ice hydrometeors (i.e. $CER_{ICE}$) decreases with increase in $N_A$

(Fig. 4b). $CER_{ICE}$ decreased from 28.1 to 23.6 µm with an increase in $N_A$ from ~35 in low aerosol loading to ~76 in high aerosol loading conditions. This micro-physical association leads to a linear positive association between $N_A$ and $A_{All-Sky}$ (Fig. 4c). $A_{All-Sky}$ increased from nearly 0.27 to 0.34 with an increase in $N_A$ from ~40 in low aerosol loading to ~80 in high aerosol loading conditions. Considering an incoming solar radiation of about 1200 W/m² at 200 hPa level, the $N_A$-associated increase in $A_{All-Sky}$ of about 0.07 can result in nearly 85 W/m² of the solar energy reflected back at the TOA. This estimate is very close to the observed increase in SWCRF$_{TOA}$ due to increase in AOD from 0.25 to 0.75 in Fig. 2c. In comparison, the corresponding OLR values illustrates an increase in $N_A$ from low to high aerosol loading scenario results in nearly 25 W/m² reduction in OLR, which is also close to the estimated LWCRF$_{TOA}$ in Fig. 2c. In agreement with these observations, reducing CCN concentration in WRF simulation also induced a systematic reduction in $CER_{ICE}$, increase in $A_{All-Sky}$ and reduced OLR irrespective of the location of the cloud top height (Supplementary Figure 12).

Figure 5 presents an illustration of how AIvE influences the cloud macro-physical, micro-physical, and radiative properties. Compared to a low aerosol loading scenario, AIvE causes more water mass aloft across the freezing level due to higher buoyancy,

and thus enhances the formation of ice-phase hydrometeors in growing TCs under a high aerosol loading scenario. These micro-physical changes subsequently intensify the updrafts and the TCs continue to develop until it reaches the tropopause. Near the tropopause, these growing TCs start expanding horizontally into thick SACs in the upwind direction. Eventually, most of the large hydrometeors fall down as precipitation at the surface and the remaining SACs stay for longer period containing relatively smaller ice particles as these particles have higher buoyancy. In contrast, under low aerosol loading conditions, the TCs mature into short-lived anvils much below the tropopause. AIvE-induced increase in ice amount of SACs and smaller hydrometeors results in significant enhancement in the SW reflectance of clouds leading to enhanced cooling (much more than the LW warming caused by an increase in cloud top). These morphological and micro-physical changes explain the observed linear relationships in Fig. 2a. Moreover, the longer lifetime of these SACs may further intensify a net cooling effect of clouds[5]. Using instantaneous mid-day satellite measurements, Peng et al.[36] illustrated that AIvE-induced net cooling effect of mixed-phase clouds is ~70 W/m$^2$ per AOD and ~15 W/m$^2$ per AOD over tropical land and ocean, respectively. Similarly, using ground-based measurements, a daytime mean net cooling effect of ~10–15 W/m$^2$ per AOD is also reported over the continental USA and China[29] using very sophisticated aerosol–cloud interactions in high-resolution WRF-Chem simulations. Nevertheless, both these studies have also found that AIvE significantly impacts the amount and optical properties of stratiform clouds under convective conditions which results in a net cooling effect at both the TOA and the surface. The magnitude of instantaneous AOD-NETCRF gradients (Fig. 2) and diurnal mean CCN-induced reduction in NETCRF (Supplementary Figure 9) are comparatively higher than previously reported values, probably because of the high pollution levels (mean AOD > 0.4) over the ISMReg. The prevalence of TEJ over the ISMReg during monsoon may further enhance the overall processes of AIvE-induced formation of SACs compared to other heavily polluted regions like China.

In the absence of the SWCRF, AIvE-induced enhancement of LWCRF results in a cloud-mediated warming effect. Thus, AIvE leads to a reduction in the maximum daytime temperature ($T_{max}$) by ~1 K (Supplementary Figure 13A), but the same increases the minimum nighttime temperature ($T_{min}$) by ~0.5 K (Supplementary Figure 13B). As a result, the simulated diurnal temperature range (DTR = $T_{max}$ − $T_{min}$) reduces by more than 1 K due to AIvE effect. Aerosol-induced higher daily minimum temperature (~0.6 K) might be contributing to the observed nighttime warming trend during the past few decades[63]. Also, AIvE-associated simultaneous surface cooling and atmospheric warming further induces reduced lower tropospheric stability, defined as the difference in temperature between model layers corresponding to ~850 hPa and surface (Supplementary Figure 13C), particularly in the regions with higher LWCRF. This indicates that AIvE-induced increase in SACs cools the surface more strongly and simultaneously stabilizes the lower atmosphere. The associated reduction in planetary boundary layer height (Supplementary Figure 13D) may further results in lower ventilation coefficient and thereby more accumulation of aerosols within PBL[64]. Accumulation of absorbing aerosols (e.g. black carbon) can further decrease the turbulence in the lower troposphere affecting moisture transport[65] as well as feedback into the spatial distribution of cloud and rainfall[7]. All of these feedbacks can suppress subsequent convective cells/shallow clouds over the ISMReg (Fig. 5).

To summarize, we demonstrate unprecedented robust signals of AIvE-associated enhancement in stratiform anvil clouds at higher altitudes with high concentrations of relatively smaller ice hydrometeors during the Indian summer monsoon. The AIvE-associated changes in cloud structure and ice-phase microphysics subsequently enhances cloud brightening and the cooling effect of clouds at both the TOA and the surface over the ISMReg. Note that the presence of uncertainties associated with satellite retrieval, meteorological co-variability, and inherent limitations of model parameterizations make it difficult to establish accurate quantitative estimates of AIvE–CRF associations in this study. Nevertheless, in a warmer and polluted environment, as expected in future, the AIvE-induced cooling effect and its feedbacks may be critical over this region of vital hydrological significance.

## Methods

**Correlation analysis**. Here, we have used 15 years (15 June to 31 October; 2002–2016) of daily measurements of AOD from NASA's MODIS-AQUA space-craft and MERRA-2, cloud properties such as CF, CTP, cloud top temperature, COT and cloud effective radius (CER) from MODIS-AQUA. The SW and LW radiation fluxes at the TOA and the surface (SURF) and all-sky albedo are obtained from Clouds and the Earth's Radiant Energy System (CERES). The vertical profiles of ice-phase cloud effective radius (CER$_{ice}$) and ice-phase water content (WC$_{ice}$) are obtained from CloudSat. CloudSat and AQUA represent two components of the NASA's A-Train satellite constellation[66] whereas MERRA-2 is an atmospheric reanalysis of the modern satellite era produced by NASA's Global Modeling and Assimilation Office (GMAO)[54]. All these measurements were extracted at a reso-lution of 1° × 1°grid over Indian summer monsoon region (17°N–27°N and 74° E–88°E, hereafter referred to as the ISMReg shown by bounded box in Supple-mentary Figure 1A). In the analysis, only the collocated data points from MODIS, MERRA-2, CERES and CloudSat measurements are obtained (corresponding to MODIS equator-crossing time over India, i.e., 13:00 local time) within the ISMReg. SW and LW CRF at both the TOA and the surface are derived from all-sky and clear-sky fluxes[67,68] using XCRF$_{TOA}$ = $X_{clr,TOA}$−$X_{all,TOA}$ and XCRF$_{SURF}$ = $X_{all,SURF}$−$X_{clr,SURF}$, where $X_{clr}$ ($X_{all}$) is the clear-sky (all-sky) SW/LW flux. Subse-quently, NETCRF is calculated by adding SWCRF and LWCRF.

**Calculation of TAR**. We used MODIS Level 2 retrieved ice-phase cloud optical depth (COT$_{ICE}$) with resolution of 1 km. Following Koren et al.[59] methodology, we used COT$_{ICE}$ of 10 to distinguish between tower (COT$_{ICE}$ > 10) and anvil (COT$_{ICE}$ < 10) cloud pixels. We then counted the number of deep convective/tower clouds and stratiform/anvil cloud pixels in each 1° × 1° grid box to calculate tower-to-anvil ratio (TAR). The TAR is defined as the number of pixels in the tower regime ($N_T$) divided by the numbers of pixels in the anvil regime ($N_A$) in each 1° × 1° grid box over the ISMReg.

**WRF model details and evaluation**. WRF has good ability in simulating the Indian monsoon depressions[69]. WRF model, version 3.6.1 is configured to simulate the regional weather prevalent over India using three domains (Supplemen-tary Fig. 5) during 12–17th August 2011. The middle domain (9 km resolution) helps to bridge the resolution difference between outer (27 km resolved) and inner most domains (3 km resolution). There are 34 vertical layers between the surface and 50 hPa. The initial and lateral boundary conditions for the meteorological fields are obtained from the NCEP-FNL data available at a resolution of 1° × 1° for every 6 h. As NCEP provided coarse initial and boundary conditions, the nested simulation with a South Asia domain at 27 km is used for better interpolation and simulation of the large-scale dynamics and physics that are fed into the inner domains as boundary conditions. A spin-up time of 1 day is considered.

MYJ boundary layer scheme[70] and NOAH land surface model[71] were used for parameterizing the planetary boundary layer and the surface energy balance, respectively. While, the inner most domain explicitly resolved clouds at 3 km resolution, Grell-Freitas cumulus parameterization scheme[72] is used in the outer and middle domain as cumulus parameterization. The updated Thompson bulk microphysics scheme with five separate species: cloud water, cloud ice, rain, snow, and a hybrid graupel with hail category[73] is used as the microphysics scheme. It incorporates the activation of water friendly aerosols (proxy for CCN) and ice nuclei (IN) and, therefore, explicitly predicts the droplet number concentration of cloud water as well as the number concentrations of the two new aerosol variables, one each for CCN and IN. A look up table is used to activate fraction of the CCN concentration into cloud droplets using prognostic temperature, vertical velocity, CCN concentration and hygroscopicity parameter and aerosol mean radius (0.04 mm). Upon nucleation, the participating aerosols are removed from the population and returned to CCN on evaporation of cloud or raindrops. Furthermore, wet scavenging is also considered in these simulations. The number of IN particles that nucleate into ice crystals is determined following DeMott et al.[74]. Further, detailed descriptions of the numerous process rate terms for cloud species, droplet number concentration, CCN, and IN can be found in Thompson et al.[73,75] and Thompson and Eidhammer[58].

Horizontal and vertical advection of winds, temperature, water vapor, cloud particles, trace gases, IN and CCN number concentrations are mixed consistently with heat, moisture, and momentum fluxes by the dynamical core of the model using a positive definite, monotonic scheme[76]. Further details about the global climatology of these aerosol species and the CCN input data are given in Thompson and Eidhammer[58]. Recent surface and aircraft measurements have reported high CCN concentration (2000–6000 #/cm$^3$) within boundary layer during monsoon season over the IGP[43,77–79]. The simulated CCN values were >2500 #/cm$^3$ over entire IGP during 12–17th August 2011 prevalent at 850 hPa altitude over Domain 3. In addition, most of the CCN burden is concentrated within boundary layer (figure is not shown). Thus, this methodology of prescribing CCN emission fluxes is able to mimic reasonably well the spatial and vertical distribution of CCN over North India during the period of study. The simulated CCN concentration is used to predict cloud water droplet concentration spectrum in the micro-physical module. Further, the interactions between clouds and radiation are implemented by linking the predicted cloud water and a constant effective radius from the microphysics scheme with the RRTMG radiation scheme[80].

**Code availability**. Weather Research Forecasting model source codes are available publicly from the website http://www2.mmm.ucar.edu/wrf/users/download/get_source.html and aerosol-sensitive cloud-resolving model simulations data will be made available upon request.

## Data availability

Satellite (MODIS and CERES) and reanalysis model (MERRA-2 and GDAS) datasets used in this study are available publicly from their respective online data archives (refer to Supplementary Table 1 in the Supplementary Information).

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

## Acknowledgements

V.P.K. would like to thank University Grants Commission, Government of India for Faculty Recharge Program Award (Ref. No. F.4-5/230-FRP/2015/BSR) and Department of Science & Technology (DST)-SERB (grant ECR/2016/001333). S.N.T. acknowledges the Earth System Science Organization, Ministry of Earth Sciences, Government of India (grant MM/NERC-MoES-03/2014/002) under the INCOMPASS campaign and the Monsoon Mission. C.S. was also partially supported by the DOE Office of Science Biological and Environmental Research (BER) Atmospheric System Research (ASR) program. The Pacific Northwest National Laboratory (PNNL) is operated for DOE by the Battelle Memorial Institute under contract DE-AC06-76RLO 1830.We also thank Puneet Sharma for helping in data extraction. Satellite (MODIS and CERES) and reanalysis (MERRA-2 and GDAS) datasets used in this study are acknowledged. All figures are produced in Interactive Data Language (IDL) 7.0.

## Author contributions

C.S., V.P.K. and S.N.T. conceived the study. C.S. and V.P.K. did data analysis, made the plots and wrote the initial manuscript. All authors have contributed to the research and revisions of the manuscript.

## Additional information

**Competing interests:** The authors declare no competing interests.

