## [Peer Review File · Nature Communications]

Reviewer #1 (Remarks to the Author):

The submitted paper “Aerosol-induced intensification of cooling effect of clouds during Indian summer monsoon” presents a study arguing that the aerosol mainly result a cooling effect of cloud radiation during summer monsoon period in Indian based on the retrievals of multi-satellite between 2002 and 2016. It focuses on a critical region of global water circling and a not fully explored factor of climate system-cloud radiation, therefore their finding can be considered as a supplement of the study of Aerosol Invigoration Effect (AlvE) on cloud radiative forcing and would draw others attentions in the community.

AlvE has been widely studied by both observations and model simulations, but not much are related to the cloud radiative effect in India, and the result of overall cooling effect is different from previous ones and interesting.

However, following questions are required to strengthen their conclusions before possible publication in the journal:

(1) The major claim of this paper is that AlvE would lead to a net cooling effect. But many previous studies propose that: The mesoscale convective clouds are already very thick, therefore, the enhancement of cloud thickness induced by AlvE would not lead to large increase of negative shortwave cloud radiative forcing (SWCRF), but the widely spread anvil would lead to large increase of positive longwave cloud radiative forcing (LWCRF), therefore, the net cloud radiative effect would be a warming effect. It is a very interest finding, but if the paper could explore deeper and come up with possible physical explanations rather than describe it as the results of continental condition, their argument would be much more strengthened.

(2) Also, previous studies show that from the physical mechanism and observations, the enhancement of cloud fraction and cloud radiative forcing by AlvE would become neutral when the AOD increase up to approximately 0.3, but the results in current paper indicate that it is not the case in India. Hope the possible explanations can be investigated and provided in the paper.

The paper proposes two main arguments that different from previously findings:

(1) The overall effect of AlvE is cooling over ISM region.

(2) AlvE would still take place when AOD is high.

Therefore, more studies should be conducted, especially modeling studies, to investigate the possible reasons behind them.

The authors should give more detail explanations about the following questions in the text:

(1) Since the study is based on the data extracted in 1×1 degree, the method of averaging for products of CloudSat should be given.

(2) In figure 2, perhaps the core figure of this paper, the AOD are divided into 200 bins, and color coded according to the sequence of the AOD, but the actual AOD values have not been provided,

which is quite important because the AOD in India should be relatively high, and under that situation, the Alve should be different from previous studies that were done in relatively clean conditions.

(3) I suppose that the “n=350” in figure 2a and 2f represent that number of samples, and the number of days since the study is based on daily data. But since the paper has used data of 3 and a half months in 15 years, it would be very interesting to see the distribution of data availability in different years and months.

(4) The qualities of figures need to be improved, some symbols are hard to see, and even the overlap of figures occurs in figure 2h and figure 2i.

Additionally, the statistical approach is very straightforward. But if there is any method to filter the data, and how, is provided, the reproduction of the work could be easier for any other researcher.

Reviewer #2 (Remarks to the Author):

This paper highlights some very intriguing relationships between aerosols and clouds. I see potential here, but some serious questions regarding aerosol growth by humidification and statistical significance must be addressed. I think these can be overcome after major revision, but it will take some work.

Lines 139-140. “collocated measurements of aerosol, cloud properties and CRF in each 10×10 grid-box”. MODIS aerosol retrievals are not possible for cloudy pixels, so the meaning of “collocated” is unclear. It must mean within the same 10×10 grid-box. Are the values averages of retrievals within each 10×10 grid-box?

Line 174. Replace “manifold” with “much larger”.

Figure 2 Legend. Descriptions for panels D and E should be switched.

Line 206. Why not also enhanced absorption of solar by particles? Analysis of clearsky fluxes is needed to backup this statement.

Line 212. Please provide a citation for the MERRA AOD analysis.

Lines 213-214. Why wouldn't MERRA AOD also be influenced by humidification? The fact that MERRA AOD is highly correlated with cloud properties (line 225) suggests that it is enhanced by humidification near clouds.

Line 226. There is no Table S2.

Pages 11-13. The modeling does not establish statistical significance of the differences. How do we know these differences aren't the random result of two simulations of an unstable atmosphere. Much more analysis establishing the statistical significance of the differences is needed.

Reviewer #3 (Remarks to the Author):

The manuscript investigates the relationship between aerosol optical depth and cloud properties/cloud radiative forcing over central India using satellite observations and reanalysis data. The results support the aerosol-cloud reinvigoration effect. The observational analysis is complemented by a WRF simulation under clean and dirty atmospheric conditions.

The findings on the role of aerosols in changing cloud properties and radiative fluxes are interesting. However, I find the manuscript rather technical and difficult to read, with a rather limited breath. Overall, it is quite hard to grasp the main findings in the context of the broader research topic. Results from the WRF simulations are also quite detached from the preceding analysis. I believe the work is more appropriate for a technical journal and I do not recommend acceptance.

Response to Reviewer #1:

The submitted paper “Aerosol-induced intensification of cooling effect of clouds during Indian summer monsoon” presents a study arguing that the aerosol mainly result a cooling effect of cloud radiation during summer monsoon period in Indian based on the retrievals of multi-satellite between 2002 and 2016. It focuses on a critical region of global water circling and a not fully explored factor of climate system-cloud radiation, therefore their finding can be considered as a supplement of the study of Aerosol Invigoration Effect (AIvE) on cloud radiative forcing and would draw others attentions in the community. AIvE has been widely studied by both observations and model simulations, but not much are related to the cloud radiative effect in India, and the result of overall cooling effect is different from previous ones and interesting. However, following questions are required to strengthen their conclusions before possible publication in the journal:

Response:

We are very thankful to the Reviewer for his/her thorough reading and constructive comments and suggestions on our manuscript. We have addressed all concerns raised by the Reviewers.

Briefly, the key inclusions and associated improvements in the revised manuscript are:

- 1) Additional observational analysis of MODIS-Level 2 data is included. Level 2 (1 km pixel) data is used to calculate the number of deep convective/tower clouds and stratiform/anvil clouds in each $1^{\circ} \times 1^{\circ}$ grid over the ISMReg. This information is analyzed along with satellite observed cloud micro-physical and optical properties to illustrate the differences in relative contribution of tower clouds and stratiform/anvil clouds to the observed net associations (aerosol-cloud radiative forcing). This analysis provides probably realistic picture that the aerosol induced invigoration effect (AIvE) leads to enhancement in both deep convective clouds as well as the stratiform/anvil clouds. The enhanced amount of stratiform/anvil clouds leads to greater cooling effect (enhancement in cloud albedo) compared to the warming effect from the increase in cloud top height of the clouds.
- 2) Detailed analysis of the WRF simulated cloud layers is added. Using the 4-D simulated cloud fraction (Time, Latitude, Longitude, and Altitude), we have computed the statistics of macro-physical (cloud top height, cloud base height and cloud thickness) and micro-physical properties (cloud effective radius of ice-phase hydrometeors) of simulated cloud layers under high CCN and low CCN scenarios. The comparison of these two simulations corroborated our observational findings based on satellite and further provided statistical robustness (sample size is of 10^3 order), indicating that the observed aerosol-cloud associations are indeed manifested by AIvE-induced changes in microphysical-dynamical coupling and not by a mere non-physical correlation.
- 3) The discussion on the influence of cloud contamination, humidification effect, and meteorological co-variability on our analysis is rewritten for clarity.
- 4) A schematic is also included to provide a physical explanation of AIvE-induced changes in cloud structure, its effect on CRF and resultant implications on lower tropospheric thermodynamics.

- 5) The entire manuscript is thoroughly checked for linguistic errors and few figures are re-created to bring better visibility as per Reviewers' suggestions, to reduce technicality and convey the findings to a broader audience.

Below, we provide a point-by-point response to the Reviewers' comments and suggestions in RED color and the associated modifications in the revised manuscript in BLUE color.

(1) The major claim of this paper is that AIvE would lead to a net cooling effect. But many previous studies propose that: The mesoscale convective clouds are already very thick, therefore, the enhancement of cloud thickness induced by AIvE would not lead to large increase of negative shortwave cloud radiative forcing (SWCRF), but the widely spread anvil would lead to large increase of positive longwave cloud radiative forcing (LWCRF), therefore, the net cloud radiative effect would be a warming effect. It is a very interest finding, but if the paper could explore deeper and come up with possible physical explanations rather than describe it as the results of continental condition, their argument would be much more strengthened.

Response:

For a developing convective cloud system, the AIvE on shortwave (SW) CRF is always negative because of the persistent enhancement of the cloud albedo. However, in the case of a mature and dissipating deep convective cloud (DCC), the cloud albedo gets saturated; thus, AIvE-induced cooling is rather insignificant. But, the AIvE-induced deep clouds with cold cloud tops emit less thermal radiation to the space. This induces large positive longwave CRF at the top of the atmosphere ($LWCRF_{TOA}$), which may cause a net warming effect of AIvE on mesoscale convective systems (MCSs) [Fan *et al.*, 2012; Koren *et al.*, 2010a; Yan *et al.*, 2014]. In contrast, AIvE associated increase in occurrence and lifetime of stratiform/anvil branch of mature MCSs and micro-physical changes such as reduction in the size of ice particles can also significantly enhance the cloud albedo, thus, causing a net cooling at the TOA and the surface [Fan *et al.*, 2013; Peng *et al.*, 2016; Stevens and Feingold, 2009]. Thus, the overall net radiative effect associated with the AIvE is a function of both macro- and micro-physical properties of the prevalent mesoscale cloud systems. Thus, in agreement with the Reviewer suggestions, we have included a detailed analysis of satellite data and model simulations to compare the impact of CCN on anvil and tower clouds, separately and explained the observed CCN-induced net cooling effect.

Additional satellite analysis using MODIS-Level 2 data is included. Level 2 (1 km pixel) data is used to calculate the number of deep convective/tower clouds and stratiform/anvil clouds at $1^\circ \times 1^\circ$ grids and daily resolution over the ISMReg. This analysis provides a better picture that the aerosol induced invigoration effect (AIvE) causes enhancement in both deep convective clouds as well as the stratiform/anvil clouds. Also, the smaller size ice hydrometeors are present in the polluted stratiform/anvil clouds and their thickness increases with aerosol loading. This enhances the lifetime and amount of stratiform clouds leading to greater cooling effect (enhancement in cloud albedo) compared to the warming effect from the increase in cloud top height. Detailed analysis of the WRF simulated cloud layers also supports this finding. Using the 4-D simulated cloud fraction (Time, Latitude, Longitude and Altitude), we computed the statistics of macro-physical (cloud top height, cloud base height and cloud thickness) and micro-physical properties (cloud effective radius of ice-phase hydrometeors) of simulated cloud layers under high CCN and low CCN scenarios. The comparison of these two simulations strengthens our observational

findings by providing statistically robust (sample size is of 10^3 order) evidence of AIvE-induced enhancement in stratiform and deep convective clouds at relatively higher altitudes.

We found that as the tower clouds deepens under AIvE and reach above 300 hPa pressure level, these deep cloud systems are advected by the prevalent Tropical Easterly Jet (TEJ). The seasonal appearance of TEJ over IMSReg plausibly amplifying the overall processes of AIvE-induced enhancement of SACs over this region compared to other continental polluted regions.

A large portion of the results and discussion part is modified with these additional analyses as shown below in blue color.

Figures 3,4 and 5, and the following text have been included in the revised manuscript:

Fig. 3. Cloud occurrence frequency as a function of tower-to-anvil ratio (TAR) and CTP for (A) low AOD and (B) high AOD. (C) MODIS observed difference (high-low AOD) in the cloud occurrence frequency as a function of TAR and CTP. (D) WRF model simulated difference (high-low CCN) in the cloud occurrence frequency as a function of cloud thickness and cloud top height for the DCC case study.

For a detailed understanding, the AIvE-induced changes in convective tower clouds (TCs) and stratiform anvil clouds (SACs) separately, and their contribution to the overall observed AOD-CRF associations are investigated. Following Koren et al. [Koren et al., 2010b], MODIS-retrieved Level-2 COT_{ICE} in each $1^\circ \times 1^\circ$ grid is used to calculate the observed number of tower (N_T : $COT_{ICE} > 10$) and anvil (N_A : $COT_{ICE} < 10$) cloud pixels (see methods for details). Daily tower-to-anvil ratio ($TAR = N_T/N_A$) within each $1^\circ \times 1^\circ$ grid over the ISMReg (collocated data point to all other datasets) is calculated. Further, all collocated cloud, AOD and radiation data are

segregated into two bins: high AOD scenario (AOD > 66 percentiles) and low AOD scenario (AOD < 33 percentiles), and OLR < 240 W/m² is used in the analysis to represent convective cloud conditions. The cloud occurrence frequency distribution for low (Fig. 3A) and high (Fig. 3B) AOD scenarios and difference in high and low AOD scenarios (Fig. 3C) are plotted on CTP–TAR space. The TAR values are arranged in ascending order and divided into 6 bins of 18 percentiles each, discarding 2 percentiles from either end to avoid extreme values. CTP bins are also found a similar way to create 36 equally populated data sample groups in the CTP–TAR space. The CTP–TAR plots synergistically illustrate TCs (TAR > 1) and the SACs (TAR < 1). It is apparent that the dominance of SACs increases as we move from right to left in the CTP–TAR space. For low aerosol loading scenario, the peak frequency of occurrence of TCs is at CTP of about 450 hPa, and the same for anvils (at constant TAR), increases from CTP of about 400 hPa (for TAR of ~0.5) to CTP of about 250 hPa (for TAR of ~0.05). But, for high aerosol loading scenario, the peak frequency of occurrence of TCs as well as anvils is at higher altitude (CTP of about 250 hPa). Further, the difference between these scenarios clearly illustrates that the occurrence of TCs and SACs increases at all altitudes above 500 hPa. Interestingly, the increase in the frequency of SACs is greater than that in DCCs under high aerosol loading. In general, most of the TCs grow and mature upto ~400 hPa, then it tends to grow vertically as they transform into anvils in the low aerosol loading scenario, perhaps due to radiative differences at the top and bottom of anvils [Hartmann and Berry, 2017]. In contrast, for high aerosol loading condition, most of the TCs fully develop into deep convective clouds due to AIvE and advect out as stratiform/anvil clouds because of the thermal capping at the tropopause and/or advective force from the tropical easterly jet (TEJ) at altitudes above 300 hPa (nearly 10 km) during ISM [George et al., 2018; Sathiyamoorthy et al., 2004].

In this framework, we also used our model simulations to compare the AIvE-induced changes in cloud distribution between low and high CCN simulations. Simulated cloud fraction profile at hourly resolution is used over each grid of the model to locate the cloud base, cloud top height (CTH), and cloud thickness (CT) of various cloud layers. A cloud layer is defined as a continuous stretch of finite cloud fraction values which exceeds more than 2 km. There may be more than one cloud layers over a grid column, but we have considered the topmost cloud layer to be consistent with MODIS CTP observations. The altitude of the model levels corresponding to cloud top and cloud base of each cloud layer is stored. CT of each cloud layer is calculated by subtracting cloud base height from the CTH. The aerosol-induced (high-low CCN) changes in cloud distribution (Fig. 3D) is plotted on CTH–CT space. The clouds lying close to the diagonal line connecting the left-bottom and right-top corner on CTH–CT axes are the growing TCs (as their CT increase almost linearly with CTH) whereas, the clouds in the right top quadrant (high CTH and high CT) represent fully developed TCs. The SACs are represented by clouds with high CTH (~12-16 km) but CT less than 4-5 km).

The simulated cloud distribution is dominated by the presence of thick cloud layers (9-11 km) at 12-16 km altitude. In addition, numerous cloud layers with CTH at 12-16 km and CT of about 3-7 km are simulated. Thus, the case study analyzed here represents well developed TCs and associated SACs over the ISMReg (Figures S11A and S11B). The aerosol-induced differences in the cloud occurrence frequency (Fig. 3D) clearly indicate that the clouds with deeper CTH and larger CT increased significantly under high CCN case indicative of the AIvE. Also, for clouds with CTH > 13 km, the increased amount of thicker SACs, but decreased thinner anvils (CT < 4 km), suggesting the formation of thicker stratiform/anvil clouds under polluted condition. It is worth mentioning that about 20 thousand cloud layers are used in this analysis, which underlines the statistical robustness of these simulated differences. While, the

environmental condition can impact the magnitude of the aerosol-induced enhancement in SACs, the fact that our model comparison is in remarkably good agreement with both MODIS and MERRA-2 analysis, strongly reinforces the seminal role of AIvE on cloud structure and distribution in MCSs over the ISMReg.

Fig. 4. Associations of A) N_T , B) CER_{ICE} , C) $A_{All-sky}$ and D) OLR with N_A as a function of MODIS AOD. Each scatter point (plus sign) in all panels (A-D) is the average of ‘n’ number of data samples ($n=57$).

Further, the associations of N_A-N_T , N_A-CER_{ICE} , $N_A-A_{All-sky}$ and N_A-OLR all as a function of AOD are studied (Fig. 4). Here, 50 scatter points are created using the same methodology as in Fig. 2. First, as expected, a linear increase in N_A and N_T is observed with an increase in aerosol loading. The ratio of N_A and N_T is nearly 1 (the bottom left corner) for low aerosol loading, but the distribution is heavily skewed towards N_A under high aerosol loading suggesting the large increase in N_A compared to N_T . Qualitatively, N_A (N_T) increased from ~ 35 (21) to ~ 76 (36) as AOD increased from 0.25 to 0.75. This emphasizes that AIvE cause significant enhancement of SACs. Interestingly, it is also seen that the size of ice hydrometeors (i.e. CER_{ICE}) decreases with increase in N_A (Fig. 4B). CER_{ICE} decreased from $28.1 \mu m$ to $23.6 \mu m$ with an increase in N_A from ~ 35 in low aerosol loading to ~ 76 in high aerosol loading conditions. This microphysical association leads to a linear positive association between N_A and $A_{All-sky}$ (Fig. 4C). $A_{All-sky}$ increased from nearly 0.27 to 0.34 with an increase in increase in N_A from ~ 40 in low aerosol loading to ~ 80 in high aerosol loading conditions. Considering an incoming solar radiation of about $1200 W/m^2$ at 200 hPa level, the N_A -associated increase in $A_{All-sky}$ of about 0.07 can result in nearly $85 W/m^2$ of the solar energy reflected back at the TOA. This estimate is very close to the observed increase in $SWCRF_{TOA}$ due to increase in AOD from 0.25 to 0.75 in Fig 2C. In comparison, the corresponding OLR values illustrates an increase in N_A from low to high aerosol loading scenario results in nearly $25 W/m^2$ reduction in OLR, which is also close to the estimated $LWCRF_{TOA}$ in Figure 2C. In agreement with these observations, reducing CCN concentration in WRF simulation also induced a systematic reduction in CER_{ICE} , increase in $A_{All-sky}$ and reduced OLR irrespective of the location of the cloud top height (Fig.S12).

Fig. 5: Conceptual schematic of processes leading to intensified cooling effect of clouds under polluted scenario over the ISMReg.

Figure 5 presents an illustration of how AIvE influences the cloud macro-physical, micro-physical and radiative properties. Compared to low aerosol loading scenario, AIvE causes more water mass aloft across the freezing level due to higher buoyancy, and thus enhances the formation of ice-phase hydrometeors in growing TCs under high aerosol loading scenario. These microphysical changes subsequently intensify the updrafts and the TCs continue to develop until it reaches the tropopause. Near the tropopause, these growing TCs start expanding horizontally into thick SACs in the upwind direction. Eventually, most of the large hydrometeors fall down as precipitation at the surface and the remaining SACs stay for longer period containing relatively smaller ice particles as these particles have higher buoyancy. In contrast, under low aerosol loading conditions, the TCs mature into short-lived anvils much below the tropopause. AIvE-induced increase in ice amount of SACs and smaller hydrometeors results in significant enhancement in the shortwave reflectance of clouds leading to enhanced cooling (much more than the LW warming caused by an increase in cloud top). These morphological and micro-physical changes explain the observed linear relationships in Fig. 2A. Moreover, the lifetime of these anvils may further intensify a net cooling effect [Stevens and Feingold, 2009]. Using instantaneous mid-day satellite measurements, Peng *et al.*, [Peng *et al.*, 2016] illustrated that AIvE-induced net cooling effect of mixed-phase clouds is $\sim 70 \text{ W/m}^2$ per AOD and $\sim 15 \text{ W/m}^2$ per AOD over tropical land and ocean, respectively. Similarly, using ground-based measurements, a daytime mean net cooling effect of $\sim 10\text{-}15 \text{ W/m}^2$ per AOD is also reported over the continental USA and China [Fan *et al.*, 2013] using very sophisticated aerosol–cloud interactions in high resolution WRF-Chem simulations. Nevertheless, both these studies have also found that AIvE significantly impacts the amount and optical properties of stratiform clouds under convective conditions which results in net cooling effect at both the TOA and the surface. The magnitude of instantaneous AOD-NETCRF gradients (Fig. 2) and diurnal mean CCN-induced reduction in NETCRF (Fig. S9) are comparatively higher than previously reported values, probably because of the high pollution levels (mean AOD > 0.4) over the ISMReg. The prevalence of TEJ over the

ISMReg during monsoon may further enhance the overall processes of AIVe-induced formation of SACs compared to other heavily polluted regions like China.

(2) Also, previous studies show that from the physical mechanism and observations, the enhancement of cloud fraction and cloud radiative forcing by AIVe would become neutral when the AOD increase up to approximately 0.3, but the results in current paper indicate that it is not the case in India. Hope the possible explanations can be investigated and provided in the paper.

Response:

Previous aerosol–cloud study by *Koren et al.* [2008] has shown that a critical AOD value (~0.3) exists over the Amazon region beyond which cloudiness decreases with AOD. Their study attributed this association to the interplay between aerosol-induced microphysical enhancements of cloudiness versus cloud inhibition due to semi-direct effect. Cloud inhibition effect depends on the ambient cloud fraction and is dominating at lower CF values (below ~0.5) because more extensive cloud cover reduces the solar radiation reaching below the cloud layers (where aerosols are present) and thus, the aerosols burn out of clouds.

Even if we keep aside the AIVe-induced enhancement in CF, availability of a large amount of moisture during monsoon season inherently results in widespread cloud coverage over ISMReg. For example, CF of ~0.75 is present corresponding to AOD ~0.4 (Fig. 2) over ISMReg. This induces a substantial reduction in the incoming solar radiation over the ISMReg [*Padma Kumari and Goswami*, 2010]. Therefore, despite the high emission rate of absorbing aerosols over the ISMR [*Bond et al.*, 2004], the aerosol-induced cloud inhibition effect seemed to have been reduced to a second-order process during the Indian summer monsoon [*Sarangi et al.*, 2017].

Nonetheless, several studies have investigated aerosol-cloud associations over India using various models, over different years and using separate datasets. Most of these studies have indicated the predominance of AIVe over ISM region [*Hazra et al.*, 2013; *Hazra et al.*, 2017; *Khain et al.*, 2013; *Pandithurai et al.*, 2012; *Sarangi et al.*, 2015; *Sarangi et al.*, 2017; *Sengupta et al.*, 2013] supporting the argument that cloud inhibition effect is negligible over ISMReg.

The paper proposes two main arguments that different from previously findings:

(1) The overall effect of AIVe is cooling over ISM region.

(2) AIVe would still take place when AOD is high.

Therefore, more studies should be conducted, especially modeling studies, to investigate the possible reasons behind them.

Response:

We truly agree with the Reviewers' point of view about the new findings in this study. As discussed in the responses above, we have indeed undergone major revision to include more detailed analysis of the reasons behind these observations.

The authors should give more detail explanations about the following questions in the text:

(1) Since the study is based on the data extracted in 1×1 degree, the method of averaging for products of CloudSat should be given.

Response:

Here, using $1^\circ \times 1^\circ$ gridded data, we first identified collocated finite aerosol (i.e. AOD from both MODIS and MERRA-2), and finite CER_{ICE} (and WC_{ICE}) and altitude sample points from CloudSat over ISMR region for the entire study period (i.e. concatenated). We have then gridded collocated AOD sample points as a function of CER_{ICE} (and WC_{ICE}) and altitude, with CER_{ICE} gridsize of $5.0 \mu m$ (and WC_{ICE} gridsize of $0.05 g/m^3$) and altitude of 0.5 km. This means that in each CER_{ICE} - altitude grid box, we have calculated mean AOD for corresponding collocated sample points.

(2) In figure 2, perhaps the core figure of this paper, the AOD are divided into 200 bins, and color coded according to the sequence of the AOD, but the actual AOD values have not been provided, which is quit important because the AOD in India should be relative high, and under that situation, the AIVe should be different from previous studies that done in relative clean condition.

Response:

We are thankful the Reviewer for pointing this out. We now show actual AOD values for both MODIS and MERRA-2 in Figure 2.

Fig.2. Top Panel: Associations of MODIS-AQUA retrieved AOD with MODIS Cloud and CERES forcing parameters. Scatter plots between A) $A_{All-Sky}$ and CF, B) CTP and CF, C) $LWCRF_{TOA}$ and $SWCRF_{TOA}$, D) $LWCRF_{SURF}$ and $SWCRF_{SURF}$ and E) $NETCRF_{TOA}$ and $NETCRF_{SURF}$, respectively, all as a function of AOD. Bottom Panel: Same as Top Panel, but for MERRA-2 AOD. Y-axis in panels C-D) and H-I) are downscaled by a factor of $1/4$. The diagonal lines in C-E) and H-J) indicate a 1:1 line. Each scatter point (plus sign) in all panels (A-J) is average of ‘n’ number of data samples ($n=1148$).

(3) I suppose that the “n=350” in figure 2a and 2f represent that number of samples, and the number of days since the study is based on daily data. But since the paper have used data of 3 and a half month in 15 years, it would be very interest to see the distribution of data availability in different year and month.

Response:

'n' has changed now since we show only 50 scatter points as opposed to 200 points earlier. Also note that 'n' is the same in all panels (A to J). To avoid repetition, we are not writing it on each panel. We explain here complete methodology adopted to create the scatter plot.

Using $1^{\circ} \times 1^{\circ}$ gridded data, we first found collocated finite aerosol (i.e. AOD from MODIS and MERRA-2), CTP, CF, LWCRF_{TOA}, LWCRF_{SURF}, SWCRF_{TOA}, SWCRF_{SURF}, NETCRF_{TOA} and NETCRF_{SURF} data samples over ISMR region for the entire study period (i.e. concatenated). For instance in Panel A, first AOD data samples are sorted in ascending order, with corresponding CF and A_{All-Sky} data samples. The entire data sample range is then divided into 50 bins (with each bin size of 2 percentiles), resulting in an equal number of data samples in each bin (i.e. n=1148). Then, the average of 'n' data samples is calculated in each bin and plotted as a scatter point in Panel A, creating 50 scatter points. In all panels, a scatter point (plus sign) is the average of n (=1148) number of collocated data samples of corresponding variables.

The following figures show the number of collocated data samples found in 3.5 months for each year.

(4) The qualities of figures need to be improved, some symbols is hard to see, and even the overlap of figures occurs in figure 2h and figure 2i. Additionally, the statistical approach is very straight forward. But if there is any method to filter the data, and how, is provided, the reproduction of the work could be easier for any other researcher.

Response:

The scatter point is shown by 'plus' sign and colored by actual AOD values. We now show only 50 scatter points to avoid possible over-plotting of points. We thank the Reviewer for pointing this out. We have taken care of over-lapping of panel edges or axis labels. Yes, the statistical approach is very simple and explained in response to minor point #3.

References

Bond, T. C., D. G. Streets, K. F. Yarber, S. M. Nelson, J.-H. Woo, and Z. Klimont (2004), A technology-based global inventory of black and organic carbon emissions from combustion, *Journal of Geophysical Research: Atmospheres*, 109(D14), n/a-n/a.

Fan, J., L. R. Leung, D. Rosenfeld, Q. Chen, Z. Li, J. Zhang, and H. Yan (2013), Microphysical effects determine macrophysical response for aerosol impacts on deep convective clouds, *Proceedings of the National Academy of Sciences*, 110(48), E4581-E4590.

Fan, J., L. R. Leung, Z. Li, H. Morrison, H. Chen, Y. Zhou, Y. Qian, and Y. Wang (2012), Aerosol impacts on clouds and precipitation in eastern China: Results from bin and bulk microphysics, *Journal of Geophysical Research: Atmospheres*, 117(D16), n/a-n/a.

George, G., C. Sarangi, S. N. Tripathi, T. Chakraborty, and A. Turner (2018), Vertical Structure and Radiative Forcing of Monsoon Clouds Over Kanpur During the 2016 INCOMPASS Field Campaign, *Journal of Geophysical Research: Atmospheres*, 123(4), 2152-2174.

Hartmann, D. L., and S. E. Berry (2017), The balanced radiative effect of tropical anvil clouds, *Journal of Geophysical Research: Atmospheres*, 122(9), 5003-5020.

Hazra, A., B. N. Goswami, and J.-P. Chen (2013), Role of Interactions between Aerosol Radiative Effect, Dynamics, and Cloud Microphysics on Transitions of Monsoon Intraseasonal Oscillations, *Journal of the Atmospheric Sciences*, 70(7), 2073-2087.

Hazra, A., H. S. Chaudhari, S. K. Saha, and S. Pokhrel (2017), Effect of cloud microphysics on Indian summer monsoon precipitating clouds: A coupled climate modeling study, *Journal of Geophysical Research: Atmospheres*, 122(7), 3786-3805.

Khain, A., T. V. Prabha, N. Benmoshe, G. Pandithurai, and M. Ovchinnikov (2013), The mechanism of first raindrops formation in deep convective clouds, *Journal of Geophysical Research: Atmospheres*, 118(16), 9123-9140.

Koren, I., G. Feingold, and L. Remer (2010a), The invigoration of deep convective clouds over the Atlantic: aerosol effect, meteorology or retrieval artifact?, *Atmos. Chem. Phys.*, 10(18), 8855-8872.

Koren, I., J. V. Martins, L. A. Remer, and H. Afargan (2008), Smoke Invigoration Versus Inhibition of Clouds over the Amazon, *Science*, 321(5891), 946-949.

Koren, I., L. A. Remer, O. Altaratz, J. V. Martins, and A. Davidi (2010b), Aerosol-induced changes of convective cloud anvils produce strong climate warming, *Atmos. Chem. Phys.*, 10, 5001-5010.

Padma Kumari, B., and B. N. Goswami (2010), Seminal role of clouds on solar dimming over the Indian monsoon region, *Geophysical Research Letters*, 37(6).

Pandithurai, G., S. Dipu, T. V. Prabha, R. S. Mahes Kumar, J. R. Kulkarni, and B. N. Goswami (2012), Aerosol effect on droplet spectral dispersion in warm continental cumuli, *Journal of Geophysical Research: Atmospheres*, 117(D16), n/a-n/a.

Peng, J., Z. Li, H. Zhang, J. Liu, and M. Cribb (2016), Systematic Changes in Cloud Radiative Forcing with Aerosol Loading for Deep Clouds in the Tropics, *Journal of the Atmospheric Sciences*, 73(1), 231-249.

Sarangi, C., S. N. Tripathi, S. Tripathi, and M. C. Barth (2015), Aerosol-cloud associations over Gangetic Basin during a typical monsoon depression event using WRF-Chem simulation, *Journal of Geophysical Research: Atmospheres*, 120(20), 9974-9995.

Sarangi, C., S. N. Tripathi, V. P. Kanawade, I. Koren, and D. S. Pai (2017), Investigation of the aerosol–cloud–rainfall association over the Indian summer monsoon region, *Atmos. Chem. Phys.*, *17*(8), 5185-5204.

Sathiyamoorthy, V., P. K. Pal, and P. C. Joshi (2004), Influence of the Upper-Tropospheric Wind Shear upon Cloud Radiative Forcing in the Asian Monsoon Region, *Journal of Climate*, *17*(14), 2725-2735.

Sengupta, K., S. Dey, and M. Sarkar (2013), Structural evolution of monsoon clouds in the Indian CTCZ, *Geophysical Research Letters*, *40*(19), 5295-5299.

Stevens, B., and G. Feingold (2009), Untangling aerosol effects on clouds and precipitation in a buffered system, *Nature*, *461*(7264), 607-613.

Yan, H., Z. Li, J. Huang, M. Cribb, and J. Liu (2014), Long-term aerosol-mediated changes in cloud radiative forcing of deep clouds at the top and bottom of the atmosphere over the Southern Great Plains, *Atmos. Chem. Phys.*, *14*(14), 7113-7124.

Response to Reviewer #2:

This paper highlights some very intriguing relationships between aerosols and clouds. I see potential here, but some serious questions regarding aerosol growth by humidification and statistical significance must be addressed. I think these can be overcome after major revision, but it will take some work.

Response:

We are very thankful to the Reviewer for his/her thorough reading and constructive comments and suggestions on our manuscript. We have addressed all concerns raised by the Reviewers.

Briefly, the key inclusions and associated improvements in the revised manuscript are:

- 1) Additional observational analysis of MODIS-Level 2 data is included. Level 2 (1 km pixel) data is used to calculate the number of deep convective/tower clouds and stratiform/anvil clouds in each $1^{\circ} \times 1^{\circ}$ grid over the ISMReg. This information is analyzed along with satellite observed cloud micro-physical and optical properties to illustrate the differences in relative contribution of tower clouds and stratiform/anvil clouds to the observed net associations (aerosol-cloud radiative forcing). This analysis provides probably realistic picture that the aerosol induced invigoration effect (AIvE) leads to enhancement in both deep convective clouds as well as the stratiform/anvil clouds. The enhanced amount of stratiform/anvil clouds leads to greater cooling effect (enhancement in cloud albedo) compared to the warming effect from the increase in cloud top height of the clouds.
- 2) Detailed analysis of the WRF simulated cloud layers is added. Using the 4-D simulated cloud fraction (Time, Latitude, Longitude, and Altitude), we have computed the statistics of macro-physical (cloud top height, cloud base height and cloud thickness) and micro-physical properties (cloud effective radius of ice-phase hydrometeors) of simulated cloud layers under high CCN and low CCN scenarios. The comparison of these two simulations corroborated our observational findings based on satellite and further provided statistical robustness (sample size is of 10^3 order), indicating that the observed aerosol-cloud associations are indeed manifested by AIvE-induced changes in microphysical-dynamical coupling and not by a mere non-physical correlation.
- 3) The discussion on the influence of cloud contamination, humidification effect, and meteorological co-variability on our analysis is rewritten for clarity.
- 4) A schematic is also included to provide a physical explanation of AIvE-induced changes in cloud structure, its effect on CRF and resultant implications on lower tropospheric thermodynamics.
- 5) The entire manuscript is thoroughly checked for linguistic errors and few figures are re-created to bring better visibility as per Reviewers' suggestions, to reduce technicality and convey the findings to a broader audience.

Below, we provide a point-by-point response to the Reviewers' comments and suggestions in RED color and the associated modifications in the revised manuscript in BLUE color.

Lines 139-140. “collocated measurements of aerosol, cloud properties and CRF in each $1^{\circ}\times 1^{\circ}$ grid-box”. MODIS aerosol retrievals are not possible for cloudy pixels, so the meaning of “collocated” is unclear. □ It must mean within the same $1^{\circ}\times 1^{\circ}$ grid-box. Are the values averages of retrievals within each $1^{\circ}\times 1^{\circ}$ grid-box?

Response:

Yes, “collocated” here meant retrievals of cloud properties and AOD within the same $1^{\circ}\times 1^{\circ}$ grid-box.

Line 174. Replace “manifold” with “much larger”.

Response:

We have replaced it.

Figure 2 Legend. Descriptions for panels D and E should be switched.

Response:

We have corrected this. Thank you.

Line 206. Why not also enhanced absorption of solar by particles? Analysis of clearsky fluxes is needed to backup this statement.

Response:

We have used All sky - clear sky fluxes to determine the CRF. In Line 206, we discussed that increase in NETCRF with AOD at the surface is higher than the same at the TOA. So the difference (NETCRF between TOA-SURF) indicates an increase in atmospheric warming due to aerosol associated changes in cloud properties.

Line 212. Please provide a citation for the MERRA AOD analysis.

Response: Thank you. We have cited two recent papers for MERRA-2 AOD [*Gelaro et al.*, 2017; *Randles et al.*, 2017].

Lines 213-214. Why wouldn't MERRA AOD also be influenced by humidification? The fact that MERRA AOD is highly correlated with cloud properties (line 225) suggests that it is enhanced by humidification near clouds.

Response:

There are two main sources of aerosol humidification effect in MODIS retrieved AOD values. First is from cloud contamination of MODIS-AOD retrieval in partially cloudy grids. Second is from enhanced size under the increase in ambient RH. But, being a reanalysis product, MERRA-2 AOD dataset is not cloud contaminated by cloudy pixels [*McCoy et al.*, 2017; *Randles et al.*, 2017]. Therefore, the persistence of the observed associations (Fig. 2) even with the use of MERRA-2 AOD dataset clearly removes the doubt about the presence of the first kind of humidification effect in our analysis. Also, we have not included AOD values >1 in our analysis to further reduce the contribution of cloud contamination in this study. Further, multiple linear regression analysis shows that the role of meteorological co-variability and thus the contribution from the second type of humidification effect is also negligible in our analysis.

In Fig.R1 blow, we have also showed the aerosol-cloud associations using both MODIS as well as MERRA-2 data but for narrow RH variability. First, the collocated data samples of $A_{All-Sky}$, CTP, AOD (MODIS and MERRA-2) and RH are identified. Then, these collocated data samples are grouped in low and high RH regimes based on RH percentile value. The low RH regime

taken as $RH < 33$ percentile value whereas high RH regime as $RH > 66$ percentile value. The positive associations between AOD and cloud physical (i.e. CTP) and radiative (i.e. $A_{All-Sky}$) properties is present for both RH regimes (Fig. R1) indicating insignificant role of RH in the associations. Moreover, in our recent study we have made exhaustive examination of the role of aerosol humidification effect on aerosol-cloud associations over ISMReg [Sarangi et al., 2017]. The study shows that aerosol humidification effect has only a negligible contribution to the observed positive associations in AOD-cloud associations over ISMReg. Thus, the positive correlation between MERRA-2 AOD and cloud properties is likely to be due to micro-physical changes and not so much due to the humidification effect.

Fig.R1. Association of MODIS AOD (Top Panel) and MERRA-2 AOD (Bottom Panel) with A) All-Sky Albedo and B) CTP for low (blue plus sign) and high (red plus sign) RH regimes. The low RH regimes taken as $RH < 33$ percentile value whereas high RH regimes as $RH > 66$ percentile value.

We have revised this discussion in the revised manuscript and included as:

In addition, the positive association between cloud properties and AOD can also be due to meteorological co-variability under cloudy conditions [Boucher and Quaas, 2013]. We have performed multiple linear regression analysis to test the co-variability of aerosol loading and meteorological parameters (e.g. temperature, relative humidity, wind shear and geopotential height) on the CF, SWCRF, LWCRF, NETCRF, and $A_{All-sky}$. The meteorological parameters are obtained from NOAA-NCEP Global Data Assimilation System (GDAS). We found that the regressed slopes between MODIS AOD and the cloud properties are much higher compared to that due to the meteorological variability (Tables S2). This suggests that variability in cloud

physical and optical properties are closely associated with variability in AOD and not by meteorological variability. Moreover, we have recently made an exhaustive examination of the role of aerosol humidification effect on aerosol-cloud associations using narrowed RH regimes and radiosondes over the ISMReg to show that aerosol humidification effect has a negligible contribution to the observed positive associations in AOD-albedo-SWCRF over ISMReg[Saranghi *et al.*, 2017].

Line 226. There is no Table S2.

Response:

Table S2 is placed in the first page of supplementary information document.

Pages 11-13. The modeling does not establish statistical significance of the differences. How do we know these differences aren't the random result of two simulations of an unstable atmosphere. Much more analysis establishing the statistical significance of the differences is needed.

Response:

In the revised manuscript, instead of using mean differences of the event (which resemble to the outcome of one event), we have used a statistical approach to analyze the CCN-sensitive WRF simulations. Using the 4-D simulated cloud fraction (Time, Latitude, Longitude and Altitude) we have computed the statistics of the macro-physical (cloud top height, cloud base height and cloud thickness) and micro-physical properties (cloud effective radius of ice-phase hydrometeors) of simulated cloud layers under high CCN and low CCN scenario. It should be noted that the number of cloud layers simulated in our model is of the order of 10 thousands and the change in the frequency of cloud occurrences between low and high CCN case is also of the order of few 1000s. Thus, a comparison of these two simulations provides a statistically robust understanding of CCN-induced changes to the cloud physical structure. The results indeed strengthen our observational findings (satellite-based) that the observed aerosol-cloud associations are manifested by AIVe-induced changes in micro-physical dynamical coupling and not by a mere non-physical correlation. As the results are a manifestation of thousands of simulated cloud layers, we believe that the observed differences are statistically significant and not a random event.

References

Boucher, O., and J. Quaas (2013), Water vapour affects both rain and aerosol optical depth, *Nature Geosci*, 6(1), 4-5.

Gelaro, R., et al. (2017), The Modern-Era Retrospective Analysis for Research and Applications, Version 2 (MERRA-2), *Journal of Climate*, 30(14), 5419-5454.

McCoy, D. T., F. A.-M. Bender, J. K. C. Mohrmann, D. L. Hartmann, R. Wood, and D. P. Grosvenor (2017), The global aerosol-cloud first indirect effect estimated using MODIS, MERRA, and AeroCom, *Journal of Geophysical Research: Atmospheres*, 122(3), 1779-1796.

Randles, C. A., et al. (2017), The MERRA-2 Aerosol Reanalysis, 1980 Onward. Part I: System Description and Data Assimilation Evaluation, *Journal of Climate*, 30(17), 6823-6850.

Sarangi, C., S. N. Tripathi, V. P. Kanawade, I. Koren, and D. S. Pai (2017), Investigation of the aerosol–cloud–rainfall association over the Indian summer monsoon region, *Atmos. Chem. Phys.*, *17*(8), 5185-5204.

Response to Reviewer #3:

The manuscript investigates the relationship between aerosol optical depth and cloud properties/cloud radiative forcing over central India using satellite observations and reanalysis data. The results support the aerosol-cloud reinvigoration effect. The observational analysis is complemented by a WRF simulation under clean and dirty atmospheric conditions.

The findings on the role of aerosols in changing cloud properties and radiative fluxes are interesting. However, I find the manuscript rather technical and difficult to read, with a rather limited breath. Overall, it is quite hard to grasp the main findings in the context of the broader research topic. Results from the WRF simulations are also quite detached from the preceding analysis. I believe the work is more appropriate for a technical journal and I do not recommend acceptance.

Response:

We are thankful to the Reviewer for appreciating the significance of our study. As per the suggestion of the Reviewer, we have rewritten a large portion of the manuscript to reduce technicality and convey the findings in a simpler way. We have included rigorous analyses to strengthen the process-level understanding of our main findings. Further, we provide conceptual illustration (Fig. 5) of the possible physical explanation of AIvE-induced changes in cloud structure, its effect on CRF and resultant implications on lower tropospheric thermodynamics. We have further analyzed the WRF simulation data using similar approach as observational findings (Figs.3 and 4) to ensure better connectivity between observational and modeling findings.

Briefly, the key inclusions and associated improvements in the revised manuscript are:

- 1) Additional observational analysis of MODIS-Level 2 data is included. Level 2 (1 km pixel) data is used to calculate the number of deep convective/tower clouds and stratiform/anvil clouds in each $1^{\circ} \times 1^{\circ}$ grid over the ISMReg. This information is analyzed along with satellite observed cloud micro-physical and optical properties to illustrate the differences in relative contribution of tower clouds and stratiform/anvil clouds to the observed net associations (aerosol-cloud radiative forcing). This analysis provides probably realistic picture that the aerosol induced invigoration effect (AIvE) leads to enhancement in both deep convective clouds as well as the stratiform/anvil clouds. The enhanced amount of stratiform/anvil clouds leads to greater cooling effect (enhancement in cloud albedo) compared to the warming effect from the increase in cloud top height of the clouds.
- 2) Detailed analysis of the WRF simulated cloud layers is added. Using the 4-D simulated cloud fraction (Time, Latitude, Longitude, and Altitude), we have computed the statistics of macro-physical (cloud top height, cloud base height and cloud thickness) and micro-physical properties (cloud effective radius of ice-phase hydrometeors) of simulated cloud layers under high CCN and low CCN scenarios. The comparison of these two simulations corroborated our observational findings based on satellite and further provided statistical robustness (sample size is of 10^3 order), indicating that the observed aerosol-cloud associations are indeed manifested by AIvE-induced changes in microphysical-dynamical coupling and not by a mere non-physical correlation.

- 3) The discussion on the influence of cloud contamination, humidification effect, and meteorological co-variability on our analysis is rewritten for clarity.
- 4) A schematic is also included to provide a physical explanation of AIvE-induced changes in cloud structure, its effect on CRF and resultant implications on lower tropospheric thermodynamics.
- 5) The entire manuscript is thoroughly checked for linguistic errors and few figures are re-created to bring better visibility as per Reviewers' suggestions, to reduce technicality and convey the findings to a broader audience.

We have included Figures 3, 4 and 5, and the following text in the revised manuscript:

Fig. 3. Cloud occurrence frequency as a function of tower-to-anvil ratio (TAR) and CTP for (A) low AOD and (B) high AOD. (C) MODIS observed difference (high-low AOD) in the cloud occurrence frequency as a function of TAR and CTP. (D) WRF model simulated difference (high-low CCN) in the cloud occurrence frequency as a function of cloud thickness and cloud top height for the DCC case study.

For a detailed understanding, the AIvE-induced changes in convective tower clouds (TCs) and stratiform anvil clouds (SACs) separately, and their contribution to the overall observed AOD-CRF associations are investigated. Following Koren et al. [Koren et al., 2010b], MODIS-retrieved Level-2 COT_{ICE} in each $1^\circ \times 1^\circ$ grid is used to calculate the observed number of tower (N_T : $COT_{ICE} > 10$) and anvil (N_A : $COT_{ICE} < 10$) cloud pixels (*see methods for details*). Daily tower-to-anvil ratio ($TAR = N_T/N_A$) within each $1^\circ \times 1^\circ$ grid over the ISMReg (collocated data

point to all other datasets) is calculated. Further, all collocated cloud, AOD and radiation data are segregated into two bins: high AOD scenario (AOD > 66 percentiles) and low AOD scenario (AOD < 33 percentiles), and OLR < 240 W/m² is used in the analysis to represent convective cloud conditions. The cloud occurrence frequency distribution for low (Fig. 3A) and high (Fig. 3B) AOD scenarios and difference in high and low AOD scenarios (Fig. 3C) are plotted on CTP–TAR space. The TAR values are arranged in ascending order and divided into 6 bins of 18 percentiles each, discarding 2 percentiles from either end to avoid extreme values. CTP bins are also found a similar way to create 36 equally populated data sample groups in the CTP–TAR space. The CTP–TAR plots synergistically illustrate TCs (TAR > 1) and the SACs (TAR < 1). It is apparent that the dominance of SACs increases as we move from right to left in the CTP–TAR space. For low aerosol loading scenario, the peak frequency of occurrence of TCs is at CTP of about 450 hPa, and the same for anvils (at constant TAR), increases from CTP of about 400 hPa (for TAR of ~0.5) to CTP of about 250 hPa (for TAR of ~0.05). But, for high aerosol loading scenario, the peak frequency of occurrence of TCs as well as anvils is at higher altitude (CTP of about 250 hPa). Further, the difference between these scenarios clearly illustrates that the occurrence of TCs and SACs increases at all altitudes above 500 hPa. Interestingly, the increase in the frequency of SACs is greater than that in DCCs under high aerosol loading. In general, most of the TCs grow and mature upto ~400 hPa, then it tends to grow vertically as they transform into anvils in the low aerosol loading scenario, perhaps due to radiative differences at the top and bottom of anvils [Hartmann and Berry, 2017]. In contrast, for high aerosol loading condition, most of the TCs fully develop into deep convective clouds due to AIVe and advect out as stratiform/anvil clouds because of the thermal capping at the tropopause and/or advective force from the tropical easterly jet (TEJ) at altitudes above 300 hPa (nearly 10 km) during ISM [George et al., 2018; Sathiyamoorthy et al., 2004].

In this framework, we also used our model simulations to compare the AIVe-induced changes in cloud distribution between low and high CCN simulations. Simulated cloud fraction profile at hourly resolution is used over each grid of the model to locate the cloud base, cloud top height (CTH), and cloud thickness (CT) of various cloud layers. A cloud layer is defined as a continuous stretch of finite cloud fraction values which exceeds more than 2 km. There may be more than one cloud layers over a grid column, but we have considered the topmost cloud layer to be consistent with MODIS CTP observations. The altitude of the model levels corresponding to cloud top and cloud base of each cloud layer is stored. CT of each cloud layer is calculated by subtracting cloud base height from the CTH. The aerosol-induced (high-low CCN) changes in cloud distribution (Fig. 3D) is plotted on CTH–CT space. The clouds lying close to the diagonal line connecting the left-bottom and right-top corner on CTH–CT axes are the growing TCs (as their CT increase almost linearly with CTH) whereas, the clouds in the right top quadrant (high CTH and high CT) represent fully developed TCs. The SACs are represented by clouds with high CTH (~12-16 km) but CT less than 4-5 km).

The simulated cloud distribution is dominated by the presence of thick cloud layers (9-11 km) at 12-16 km altitude. In addition, numerous cloud layers with CTH at 12-16 km and CT of about 3-7 km are simulated. Thus, the case study analyzed here represents well developed TCs and associated SACs over the ISMReg (Figures S11A and S11B). The aerosol-induced differences in the cloud occurrence frequency (Fig. 3D) clearly indicate that the clouds with deeper CTH and larger CT increased significantly under high CCN case indicative of the AIVe. Also, for clouds with CTH > 13 km, the increased amount of thicker SACs, but decreased thinner anvils (CT < 4 km), suggesting the formation of thicker stratiform/anvil clouds under polluted condition. It is worth mentioning that about 20 thousand cloud layers are used in this

analysis, which underlines the statistical robustness of these simulated differences. While, the environmental condition can impact the magnitude of the aerosol-induced enhancement in SACs, the fact that our model comparison is in remarkably good agreement with both MODIS and MERRA-2 analysis, strongly reinforces the seminal role of AIVe on cloud structure and distribution in MCSs over the ISMReg.

Fig. 4. Associations of A) N_T , B) CER_{ICE} , C) $A_{All-Sky}$ and D) OLR with N_A as a function of MODIS AOD. Each scatter point (plus sign) in all panels (A-D) is the average of ‘n’ number of data samples ($n=57$).

Further, the associations of N_A-N_T , N_A-CER_{ICE} , $N_A-A_{All-Sky}$ and N_A-OLR all as a function of AOD are studied (Fig. 4). Here, 50 scatter points are created using the same methodology as in Fig. 2. First, as expected, a linear increase in N_A and N_T is observed with an increase in aerosol loading. The ratio of N_A and N_T is nearly 1 (the bottom left corner) for low aerosol loading, but the distribution is heavily skewed towards N_A under high aerosol loading suggesting the large increase in N_A compared to N_T . Qualitatively, N_A (N_T) increased from ~35 (21) to ~76 (36) as AOD increased from 0.25 to 0.75. This emphasizes that AIVe cause significant enhancement of SACs. Interestingly, it is also seen that the size of ice hydrometeors (i.e. CER_{ICE}) decreases with increase in N_A (Fig. 4B). CER_{ICE} decreased from 28.1 μm to 23.6 μm with an increase in N_A from ~35 in low aerosol loading to ~76 in high aerosol loading conditions. This microphysical association leads to a linear positive association between N_A and $A_{All-Sky}$ (Fig. 4C). $A_{All-Sky}$ increased from nearly 0.27 to 0.34 with an increase in increase in N_A from ~40 in low aerosol loading to ~80 in high aerosol loading conditions. Considering an incoming solar radiation of about 1200 W/m^2 at 200 hPa level, the N_A -associated increase in $A_{All-Sky}$ of about 0.07 can result in nearly 85 W/m^2 of the solar energy reflected back at the TOA. This estimate is very close to the observed increase in $SWCRF_{TOA}$ due to increase in AOD from 0.25 to 0.75 in Fig 2C. In comparison, the corresponding OLR values illustrates an increase in N_A from low to high aerosol loading scenario results in nearly 25 W/m^2 reduction in OLR, which is also close to the estimated $LWCRF_{TOA}$ in Figure 2C. In agreement with these observations, reducing CCN concentration in WRF simulation also induced a systematic reduction in CER_{ICE} , increase in $A_{All-Sky}$ and reduced OLR irrespective of the location of the cloud top height (Fig.S12).

Fig. 5: Conceptual schematic of processes leading to intensified cooling effect of clouds under polluted scenario over the ISMReg.

Figure 5 presents an illustration of how AIvE influences the cloud macro-physical, micro-physical and radiative properties. Compared to low aerosol loading scenario, AIvE causes more water mass aloft across the freezing level due to higher buoyancy, and thus enhances the formation of ice-phase hydrometeors in growing TCs under high aerosol loading scenario. These microphysical changes subsequently intensify the updrafts and the TCs continue to develop until it reaches the tropopause. Near the tropopause, these growing TCs start expanding horizontally into thick SACs in the upwind direction. Eventually, most of the large hydrometeors fall down as precipitation at the surface and the remaining SACs stay for longer period containing relatively smaller ice particles as these particles have higher buoyancy. In contrast, under low aerosol loading conditions, the TCs mature into short-lived anvils much below the tropopause. AIvE-induced increase in ice amount of SACs and smaller hydrometeors results in significant enhancement in the shortwave reflectance of clouds leading to enhanced cooling (much more than the LW warming caused by an increase in cloud top). These morphological and micro-physical changes explain the observed linear relationships in Fig. 2A. Moreover, the lifetime of these anvils may further intensify a net cooling effect [Stevens and Feingold, 2009]. Using instantaneous mid-day satellite measurements, Peng *et al.*, [Peng *et al.*, 2016] illustrated that AIvE-induced net cooling effect of mixed-phase clouds is $\sim 70 \text{ W/m}^2$ per AOD and $\sim 15 \text{ W/m}^2$ per AOD over tropical land and ocean, respectively. Similarly, using ground-based measurements, a daytime mean net cooling effect of $\sim 10\text{-}15 \text{ W/m}^2$ per AOD is also reported over the continental USA and China [Fan *et al.*, 2013] using very sophisticated aerosol–cloud interactions in high resolution WRF-Chem simulations. Nevertheless, both these studies have also found that AIvE significantly impacts the amount and optical properties of stratiform clouds under convective conditions which results in net cooling effect at both the TOA and the surface. The magnitude of instantaneous AOD-NETCRF gradients (Fig. 2) and diurnal mean CCN-induced reduction in NETCRF (Fig. S9) are comparatively higher than previously reported values, probably because of the high pollution levels (mean AOD > 0.4) over the ISMReg. The prevalence of TEJ over the

ISMReg during monsoon may further enhance the overall processes of AIVe-induced formation of SACs compared to other heavily polluted regions like China.

References

Fan, J., L. R. Leung, D. Rosenfeld, Q. Chen, Z. Li, J. Zhang, and H. Yan (2013), Microphysical effects determine macrophysical response for aerosol impacts on deep convective clouds, *Proceedings of the National Academy of Sciences*, *110*(48), E4581-E4590.

George, G., C. Sarangi, S. N. Tripathi, T. Chakraborty, and A. Turner (2018), Vertical Structure and Radiative Forcing of Monsoon Clouds Over Kanpur During the 2016 INCOMPASS Field Campaign, *Journal of Geophysical Research: Atmospheres*, *123*(4), 2152-2174.

Hartmann, D. L., and S. E. Berry (2017), The balanced radiative effect of tropical anvil clouds, *Journal of Geophysical Research: Atmospheres*, *122*(9), 5003-5020.

Koren, I., L. A. Remer, O. Altaratz, J. V. Martins, and A. Davidi (2010), Aerosol-induced changes of convective cloud anvils produce strong climate warming, *Atmos. Chem. Phys.*, *10*, 5001-5010.

Peng, J., Z. Li, H. Zhang, J. Liu, and M. Cribb (2016), Systematic Changes in Cloud Radiative Forcing with Aerosol Loading for Deep Clouds in the Tropics, *Journal of the Atmospheric Sciences*, *73*(1), 231-249.

Sathiyamoorthy, V., P. K. Pal, and P. C. Joshi (2004), Influence of the Upper-Tropospheric Wind Shear upon Cloud Radiative Forcing in the Asian Monsoon Region, *Journal of Climate*, *17*(14), 2725-2735.

Stevens, B., and G. Feingold (2009), Untangling aerosol effects on clouds and precipitation in a buffered system, *Nature*, *461*(7264), 607-613.

Reviewer #1 (Remarks to the Author):

The authors have conducted plentiful additional analysis that basically good enough to answer all my concerns. Therefore, I suggest that the paper can be considered for the publication after a careful check for possible clerical errors.